# Structure of the germline genome of *Tetrahymena thermophila* and relationship to the massively rearranged somatic genome

Eileen P Hamilton[1†], Aurélie Kapusta[2†], Piroska E Huvos[3], Shelby L Bidwell[4], Nikhat Zafar[4], Haibao Tang[4], Michalis Hadjithomas[4], Vivek Krishnakumar[4], Jonathan H Badger[4], Elisabet V Caler[4], Carsten Russ[5], Qiandong Zeng[5], Lin Fan[5], Joshua Z Levin[5], Terrance Shea[5], Sarah K Young[5], Ryan Hegarty[5], Riza Daza[5], Sharvari Gujja[5], Jennifer R Wortman[5], Bruce W Birren[5], Chad Nusbaum[5], Jainy Thomas[2], Clayton M Carey[2], Ellen J Pritham[2], Cédric Feschotte[2], Tomoko Noto[6‡], Kazufumi Mochizuki[6‡], Romeo Papazyan[7], Sean D Taverna[7], Paul H Dear[8], Donna M Cassidy-Hanley[9], Jie Xiong[10], Wei Miao[10], Eduardo Orias[1], Robert S Coyne[4]*

[1]Department of Molecular, Cellular, and Developmental Biology, University of California, Santa Barbara, Santa Barbara, United States; [2]Department of Human Genetics, University of Utah School of Medicine, Salt Lake City, United States; [3]Biochemistry and Molecular Biology, Southern Illinois University, Carbondale, United States; [4]J. Craig Venter Institute, Rockville, United States; [5]Eli and Edythe L. Broad Institute of Harvard and MIT, Cambridge, United States; [6]Institute of Molecular Biotechnology, Vienna, Austria; [7]Department of Pharmacology and Molecular Sciences, The Johns Hopkins University School of Medicine, Baltimore, United States; [8]MRC Laboratory of Molecular Biology, Cambridge, United Kingdom; [9]Department of Microbiology and Immunology, Cornell University, Ithaca, United States; [10]Institute of Hydrobiology, Chinese Academy of Sciences, Wuhan, China

**\*For correspondence:** rcoyne@jcvi.org

[†]These authors contributed equally to this work

**Present address:** [‡]Institute of Human Genetics - CNRS, Montpellier, France

**Competing interests:** The authors declare that no competing interests exist.

**Abstract** The germline genome of the binucleated ciliate *Tetrahymena thermophila* undergoes programmed chromosome breakage and massive DNA elimination to generate the somatic genome. Here, we present a complete sequence assembly of the germline genome and analyze multiple features of its structure and its relationship to the somatic genome, shedding light on the mechanisms of genome rearrangement as well as the evolutionary history of this remarkable germline/soma differentiation. Our results strengthen the notion that a complex, dynamic, and ongoing interplay between mobile DNA elements and the host genome have shaped *Tetrahymena* chromosome structure, locally and globally. Non-standard outcomes of rearrangement events, including the generation of short-lived somatic chromosomes and excision of DNA interrupting protein-coding regions, may represent novel forms of developmental gene regulation. We also compare *Tetrahymena*'s germline/soma differentiation to that of other characterized ciliates, illustrating the wide diversity of adaptations that have occurred within this phylum.

## Introduction

The establishment of distinct genomic lineages (cellular or nuclear) in the life cycles of phylogenetically diverse organisms has allowed the evolution of a wide variety of programmed, somatic lineage-specific DNA rearrangement mechanisms. Some cases mediate the generation of protein products specific to a differentiated cell type, such as sigmaK of the *Bacillus subtilis* mother cell (*Kunkel et al., 1990*) or the vast diversity of vertebrate immunoglobulins (*Schatz, 2004*). Other cases result in genome-wide chromosome restructuring, as was first recognized by microscopic observation of parasitic nematodes over 125 years ago (*Boveri, 1887*) and since documented in several eukaryotic branches, including vertebrates (*Bachmann-Waldmann et al., 2004*; *Smith et al., 2012*; *Sun et al., 2014*; *Wang and Davis, 2014*). This large-scale phenomenon has been most thoroughly studied in the phylum Ciliophora, or ciliates, a deep-branching and diverse group of protozoa (*Bracht et al., 2013*; *Chalker and Yao, 2011*; *Coyne et al., 2012*; *Vogt et al., 2013*; *Yao et al., 2014*). Although unicellular, ciliates carry two distinct nuclei that display a remarkable form of germline/soma differentiation (*Figure 1A*; *Orias et al., 2011*); the smaller, diploid, transcriptionally silent germline nucleus (micronucleus or MIC) contains the genetic material transmitted across sexual generations, whereas the larger, polyploid, actively expressed somatic nucleus (macronucleus or MAC) supports all the vegetative functions of the cell. Despite differing in several fundamental features of eukaryotic nuclei, the MAC is derived from a mitotic sibling of the MIC during sexual reproduction in a process that involves extensive, genome-wide programmed DNA rearrangements.

The extent and nature of ciliate genome rearrangement vary widely within the phylum, but the two main events are chromosome fragmentation and DNA elimination (*Figure 1B*). In the widely studied model organism, *Tetrahymena thermophila*, the five MIC chromosomes are fragmented at sites of the 15 bp Chromosome breakage sequence (Cbs) (*Yao et al., 1990*) into about 200 MAC chromosomes (*Eisen et al., 2006*). Other characterized ciliates also undergo extensive chromosome fragmentation but do not display a conserved cis-acting breakage signal. It has been suggested that the evolutionary advantage of chromosome fragmentation may relate to the high ploidy of MACs (~45N for all but one chromosome in *Tetrahymena*, ~800 N in *Paramecium*, ~2000 N in *Oxytricha*) and their amitotic division mechanism, which could damage larger chromosomes or be physically constrained by their entanglement (*Coyne et al., 1996*). This amitotic mechanism also results in unequal chromosome segregation, which can lead to the generation of phenotypic diversity among the vegetative descendants of a single cell ('phenotypic assortment', documented in *Tetrahymena* (*Orias and Flacks, 1975*). In addition, fragmentation permits differential copy number control (observed in *Tetrahymena* (reviewed in *Yao et al. [1979]*), *Oxytricha* and other ciliates (*Baird and Klobutcher, 1991*; *Steinbruck, 1983*; *Swart et al., 2013*).

Concomitantly with fragmentation, thousands of Internal Eliminated Sequences (IESs; first described in *Tetrahymena* [*Yao et al., 1984*]) are spliced from the *Tetrahymena* MIC genome. In *Paramecium tetraurelia*, a fellow oligohymenophorean ciliate distantly related to *Tetrahymena* (*Baroin-Tourancheau et al., 1992*), partial assembly of the MIC genome has revealed the presence of about 45,000 short, unique copy IESs, many lying within the MIC progenitors of MAC genes (*Arnaiz et al., 2012*). The more distantly related spirotrichous ciliate, *Oxytricha trifallax* undergoes an extreme type of genome rearrangement. Roughly 16,000 MAC chromosomes (most carrying only a single gene) (*Swart et al., 2013*) are derived from a MIC genome ten times the size of the MAC genome, in a process that also involves extensive 'unscrambling' of non-contiguous MIC genome sequences (*Chen et al., 2014*).

A leitmotif of programmed genome rearrangements in many organisms is the involvement of mobile DNA elements. In some cases, this involvement is as an agent of the event, through domesticated gene products (e.g. Rag recombinases [*Fugmann, 2010*; *Jones and Gellert, 2004*; *Kapitonov and Koonin, 2015*], HO endonuclease [*Koufopanou and Burt, 2005*]); in other cases, mobile elements are a target of programmed rearrangement events (e.g. the *B. subtilis* Skin element that interrupts the sigK gene [*Takemaru et al., 1995*]). It has long been recognized that many ciliate IESs contain transposable elements (TEs) and/or their remnants and hypothesized that their elimination is a form of MAC genome self-defense (*Klobutcher and Herrick, 1997*). In both *Tetrahymena* and *Paramecium*, IES elimination requires the action of proteins domesticated from piggyBac transposases (*Baudry et al., 2009*; *Cheng et al., 2010*; *Shieh and Chalker, 2013*), as well as proteins and histone modifications associated with epigenetic TE silencing in other organisms (*Chalker et al.,*

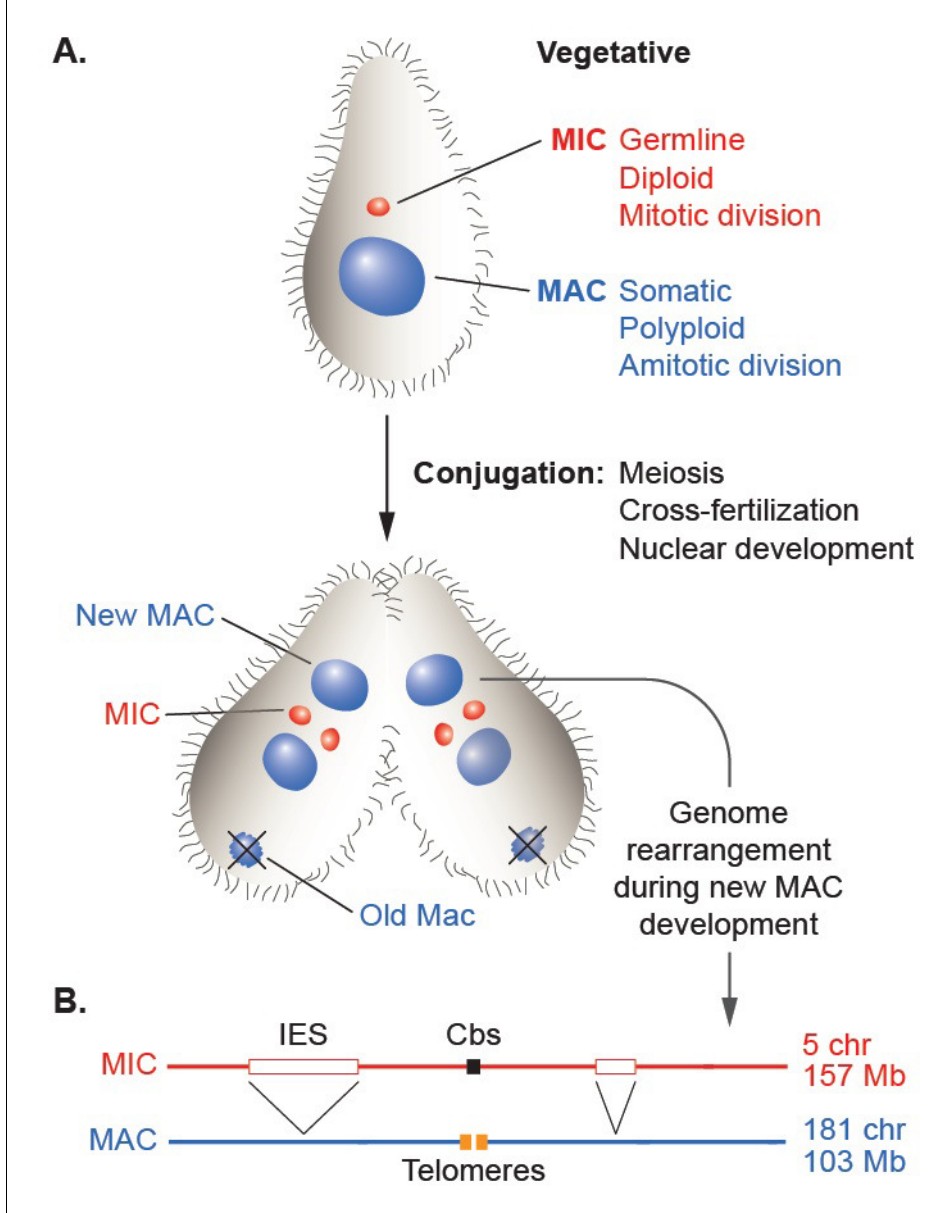

**Figure 1.** Nuclear dualism and genome rearrangement in *Tetrahymena*. (**A**) Schematic of two stages of *Tetrahymena* life cycle showing major characteristics of micronuclei (MIC; red) and macronuclei (MAC; blue) and nuclear events of conjugation. (**B**) Main events of programmed genome rearrangement. A portion of the MIC genome is shown in red, with internal eliminated sequences (IES) shown as open boxes and the Cbs sequence in black. The corresponding MAC regions (blue) lack the IESs, with the flanking MAC-destined sequences (MDSs) joined (represented by ˆ symbols). Breakage and addition of telomeres (orange boxes) has occurred at the former site of the Cbs.

The following figure supplement is available for figure 1:

**Figure supplement 1.** Tiling method used to extend scaffolds into super-assemblies.

---

*2013*). In *Oxytricha*, germline-limited transposons mediate their own excision and also contribute to other programmed rearrangement events (*Nowacki et al., 2009*). The evolutionary origins of chromosome fragmentation are less clear, but, at least in *Tetrahymena*, features of Cbs suggest a possible link to mobile elements ([*Ashlock et al., 2016*; *Fan and Yao, 2000*; *Hamilton et al., 2006b*] and

this study). Thus, the study of programmed DNA rearrangement in ciliates may help shed light on the delicate evolutionary balance that exists between mobile elements and the genomes they occupy.

Despite germline sequencing efforts in three model ciliates, *Tetrahymena* (*Fass et al., 2011*), *Paramecium* (*Arnaiz et al., 2012*), and *Oxytricha* (*Chen et al., 2014*), there is no complete picture of the architectural relationship between ciliate germline and somatic genomes. Here, we report the sequencing, assembly, and analysis of the 157 Mb MIC genome of *T. thermophila* strain SB210, the same strain whose 103 Mb MAC genome sequence we have previously characterized (*Coyne et al., 2008*; *Eisen et al., 2006*; *Hamilton et al., 2006a*). We constructed full-length super-assemblies of all five MIC chromosomes, providing a unique resource for ciliate genome analysis. By mapping a set of germline deletions against these super-assemblies, we delimited the locations of the five MIC centromeres. We mapped 225 instances of the Cbs, which define the ends of all 181 stably maintained MAC chromosomes as well as several short-lived, 'Non-Maintained Chromosomes' (NMCs), some of which contain a number of active genes. Additionally, we report multiple cases of short and long-range Cbs duplications in *T. thermophila* and the conservation of Cbs sequence and location in three other *Tetrahymena* species. We showed that approximately one third (54 Mb) of the MIC genome is eliminated in the form of around 12,000 IESs, and mapped the precise locations of over 7500, revealing their enrichment at the centers and ends of MIC chromosomes. Our comparative analysis of MIC-limited TEs shows that the majority are related to DNA (Class 2) transposons from a variety of families and suggests multiple invasions of the genome and potentially recent transpositional activity. We analyzed IES junctions and excision variability genome-wide, greatly extending previous reports of their imprecision (e.g. [*Austerberry et al., 1989*; *Li and Pearlman, 1996*; *Wells et al., 1994*]), and yet we also report a very limited number of unusual, precisely excised IESs that interrupt protein-coding regions. Our results provide the first genome-wide picture of programmed DNA rearrangements in *T. thermophila*, and support a view of the germline genome as a complex and dynamic entity, on both developmental and evolutionary timescales.

## Results and discussion

### Germline chromosome structure

#### MIC genome sequencing and chromosome-length assembly

Shotgun sequencing and assembly of the *T. thermophila* MIC genome is described in 'Materials and methods', and statistics are summarized in *Supplementary file 1A*. The final assembly is 157 Mb in length and composed of 1464 scaffolds, whereas the MAC genome assembly is 103 Mb and contains 1158 scaffolds. To fully understand the inter-relationship of the MAC and MIC genomes, it is essential to join the scaffolds of each separate assembly into complete MAC and MIC chromosomes. Extensive genome closure and HAPPY mapping efforts have produced super-assemblies of every MAC chromosome ([*Coyne et al., 2008*; *Hamilton et al., 2006a*]; *Supplementary file 1B*) but considerable uncertainty remains as to scaffold placement and/or orientation on several chromosomes. Likewise, although genetic mapping can assign some MAC chromosomes/scaffolds to locations on one of the five MIC chromosomes, their order and orientation can be hard to determine. By a MIC-MAC cross-alignment 'tiling' method (described in Materials and methods and *Figure 1—figure supplement 1*), we used each assembly to improve the other. By this process, most of the larger MIC scaffolds were linked into five chromosome-length super-assemblies that together incorporate 152 Mb of the total 157 Mb MIC assembly (*Supplementary file 1C,D*; also see 'MIC-scaff' and corresponding 'MAC-scaff' schematic concatenations in *Figure 2*). While the super-assemblies are admittedly not perfect, their uncertainties are on a small scale, and thus the maps allow observations of general trends in MIC chromosome architecture. To our knowledge, these are the first assemblies of nearly full-length ciliate MIC chromosomes and thus represent novel resources for genomic analyses. We have incorporated them into a browser (http://www.jcvi.org/jbrowse/?data=tta2mic) that relates the MIC and MAC genomes and includes many other features described below.

#### MIC centromeres

Centromeric loci play essential, highly conserved roles in the faithful segregation of chromosomes during meiosis and mitosis (*Bloom, 2014*). Recent studies (*Plohl et al., 2014*; *Topp and Dawe,*

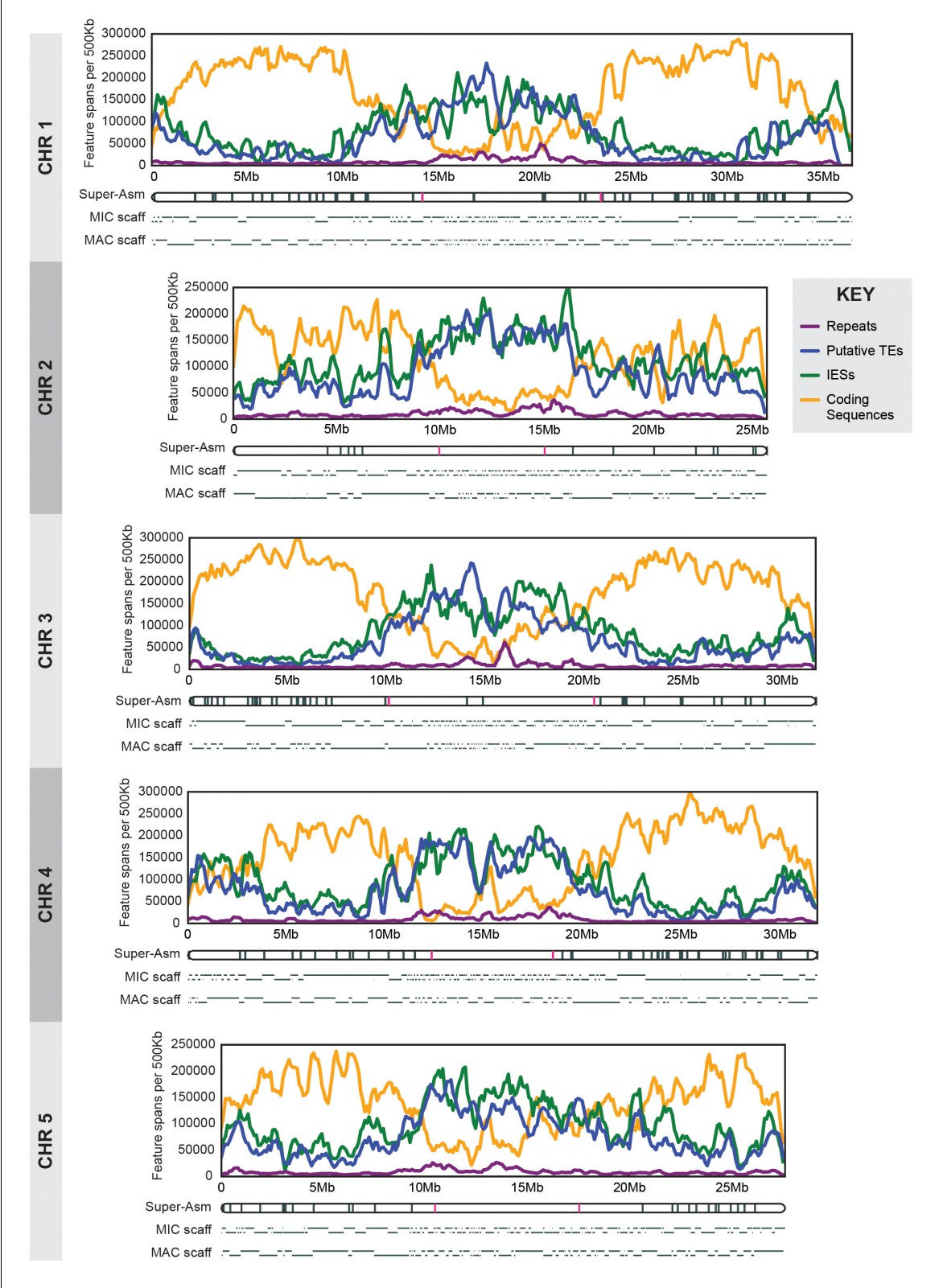

**Figure 2.** MIC chromosome landscapes. For each chromosome, the top panel shows the density of several genomic features, measured as number of base pairs (span) per 500 kb sliding window (100 kb slide increment). Purple = simple sequence repetitive DNA (note that exclusion of those simple sequence repeats that overlap with TEs has minimal effect on the distribution pattern). Blue = putative TEs. Green = high-confidence IESs. Orange = protein-coding sequences. The corresponding chromosome-length super-assembly (Super-Asm) is shown immediately below, each Cbs indicated by a

*Figure 2 continued on next page*

*Figure 2 continued*

vertical tick. Red ticks indicate Cbs's flanking putative centromeres (see main text and *Figure 2—figure supplement 1*). In the 'MIC-scaff' schematic, the scaffolds comprising each MIC chromosome super-assembly are depicted as horizontal lines (alternating in vertical position to delineate each from its neighbors). The 'MAC-scaff' schematic indicates the positions of MAC scaffolds (many of which are complete, fully sequenced MAC chromosomes) derived from the corresponding regions of the MIC chromosome. Note that, because IESs are absent from MAC scaffolds, their lengths are actually shorter, but for simplicity of viewing, these lengths have been stretched so that MAC-scaff endpoints line up with their corresponding positions in the MIC. Chromosomes are stacked so that their centers align vertically.

The following figure supplement is available for figure 2:

**Figure supplement 1.** Deletion mapping of *Tetrahymena* centromeres.

*2006*) have greatly increased understanding of centromere structure and function, but much still remains unclear. Several biological features of *Tetrahymena*, as well as its powerful experimental toolbox, have made this organism a useful model for studies of centromeric heterochromatin (*Cervantes et al., 2006*; *Cui and Gorovsky, 2006*; *Papazyan et al., 2014*), recombination (*Lukaszewicz et al., 2013*; *Shodhan et al., 2014*), chromosome cohesion (*Howard-Till et al., 2013*), and centromere evolution (*Elde et al., 2011*; *Malik and Henikoff, 2002*, *2009*), all of which would benefit from better genetic and molecular definition of its centromeres. The full-length chromosomal super-assemblies described above make this possible.

We demarcated *Tetrahymena* centromeric regions using germline, mitotically stable, chromosomal deletions isolated in a separate study (Cassidy-Hanley et al.; manuscript in preparation). Each deletion was mapped in relation to chromosome breakage sites along the length of each MIC chromosome ([*Figure 2—figure supplement 1*]; [*Cassidy-Hanley et al., 1994*]). We observed that chromosome arm deletions never extend into the central regions of MIC chromosome super-assemblies, presumably because they are essential for centromere function. Operationally (because of how the deletions were mapped), two unique Cbs's flank each putative centromere region (see red hash marks in *Figure 2*). Cytologically, all five *Tetrahymena* MIC chromosomes appear metacentric and, as expected, the midpoints of the chromosomal super-assembly lie near the centromeric region midpoints (*Table 1*). We also note, as described in *Supplementary file 1E*, that MAC chromosomes derived from MIC centromeric regions tend to be unusually large. The five putative centromeric regions range between 5.0 and 10.3 Mb and together comprise 37.8 Mb, or 24.7% of the assembled MIC genome. These estimates are subject to change in either direction for the following reasons. The centromere regions of the MIC assembly are highly fragmented (*Table 1*, column 5; *Figure 2*); missing sequence would increase their size. On the other hand, the precise endpoints of the deletions are unknown, and the complete region between flanking Cbs's may not be required for centromere function.

**Table 1.** MIC centromere regions and centric MAC chromosomes.

| MIC chromosome | L-Cbs location (Mb) | R-Cbs location (Mb) | Cen length (Mb) | # super-contigs in Cen | MIC chromosome length (Mb) | Cen midpoint (Mb) | Chromosome mid-point (Mb) |
|---|---|---|---|---|---|---|---|
| 1 | 13.98 | 23.24 | 9.26 | 87 | 36.32 | 18.61 | 18.16 |
| 2 | 9.81 | 14.85 | 5.04 | 77 | 25.51 | 12.33 | 12.76 |
| 3 | 9.98 | 20.32 | 10.34 | 120 | 31.52 | 15.15 | 15.76 |
| 4 | 12.23 | 18.34 | 6.11 | 74 | 31.72 | 15.29 | 15.86 |
| 5 | 10.37 | 17.39 | 7.02 | 62 | 27.47 | 13.88 | 13.74 |
| Total | | | 37.77 (24.7%) | | 152.54 | | |

L-Cbs and R-Cbs represent the most Cen-proximal Cbs on the left and right chromosome arms, respectively. Centromere locations were established by deletion mapping (see text for details). For chromosomes 2, 4, and 5, the L-1 and R-1 Cbs flank the putative centromere region. The remaining centromeres contain Cbs's. Cbs 3L-3 and 3R-1 flank the chromosome 3 centromere, while Cbs 1L-6 and Cbs 1R-11 flank the centromere region of chromosome 1. Locations in Mb use the far (telomere) end of the left arm as the origin.

Centromeric and pericentromeric regions generally contain repetitive sequences, often consisting of large arrays of tandem repeats interspersed with transposable elements (TEs) (*Buscaino et al., 2010*; *Hayden and Willard, 2012*; *López-Flores and Garrido-Ramos, 2012*; *Plohl et al., 2014*). We plotted the densities along each MIC chromosome of both simple sequence repeats (*Figure 2*, purple lines) and putative TEs and their remnants (blue lines; see below for a description of TE characterization) and found that both types of repetitive sequence are more prevalent in the putative centromeric regions than in the chromosome arms. These observations of large, repeat-rich centromeric regions are consistent with the 'meiotic drive' hypothesis (*Elde et al., 2011*; *Malik and Henikoff, 2002*, *2009*)—that in organisms, such as *Tetrahymena*, that undergo exclusively female meiosis (in which only one of the four meiotic products becomes a gamete), competition between sister chromosomes for transmission during meiosis will result in rapid evolution and expansion of centromeric sequences.

During formation of a new MAC in *Tetrahymena*, the centromeric histone H3 disappears from differentiating MACs, suggesting the programmed elimination of Cen-specific sequences (*Cervantes et al., 2006*; *Cui and Gorovsky, 2006*). The close, linear packing of MAC chromosome precursors along the entire length of MIC chromosomes and the presence of retained, macronuclear-destined sequences (MDSs) interspersed throughout the *Tetrahymena* centromere regions suggests that IES removal is sufficient to account for this centromere loss. In *Paramecium*, IESs found in MIC regions that give rise to MAC chromosomes are generally very short and non-repetitive (*Arnaiz et al., 2012*), thus not resembling typical centromeric DNA. However, these regions are separated by large (and as yet unassembled) blocks of repetitive DNA (*Arnaiz et al., 2012*; *Le Mouël et al., 2003*), which seem more likely to represent centromeres. Centromeric histone H3 also disappears during MAC differentiation in *Paramecium*, and this disappearance is dependent on factors required for IES excision (*Lhuillier-Akakpo et al., 2016*), suggesting that the centromeres of both organisms, despite their apparent dissimilarities, are eliminated as IESs.

## Chromosome fragmentation

In contrast to most eukaryotes, programmed somatic chromosome breakage and de novo telomere addition are part of the normal life cycles of several groups, including ciliates (*Coyne et al., 1996*) and certain parasitic nematodes (*Müller and Tobler, 2000*). Among these organisms, many details of the process differ markedly (*Amar, 1994*; *Baird and Klobutcher, 1989*; *Caron, 1992*; *Duret et al., 2008*; *Forney and Blackburn, 1988*; *Herrick et al., 1987*; *Le Mouël et al., 2003*; *Scott et al., 1993*). *Tetrahymena* carries out chromosome breakage and telomere addition with high specificity and reliability. In *T. thermophila* and related species (*Coyne and Yao, 1996*), these processes are driven by the necessary and sufficient cis-acting DNA element, Cbs (Chromosome breakage sequence), a highly conserved 15-mer (*Fan and Yao, 2000*; *Hamilton et al., 2006b*; *Yao et al., 1990*). De novo telomere addition by telomerase occurs within a region ~5–25 bp on each side of a Cbs (*Fan and Yao, 1996*); the Cbs itself and its immediate flanking sequences are found only in the MIC. Thanks to our chromosome super-assemblies, we can now investigate chromosome breakage throughout the entire *T. thermophila* genome.

### The chromosome breakage sequence (Cbs) family

We identified 225 Cbs's in the MIC genome assembly (*Supplementary file 2A*), including those associated with the ends of every MAC chromosome (*Supplementary file 2B*); thus, the Cbs family is responsible for all developmentally programmed chromosome breakage in *T. thermophila*. Positioning this complete set of breakage signals on the MIC chromosome super-assemblies makes *T. thermophila* the first ciliate in which the complete linear relationship between MIC and MAC chromosomes has been defined (see 'Super-Asm' schematic in *Figure 2*). As expected, the majority of MAC chromosomes are generated by cleavage at Cbs's that are consecutively spaced along MIC chromosomes. However, we identified seven complex MAC chromosomes that are generated not simply by conventional fragmentation, but also by the site-specific joining of non-contiguous segments of germline DNA. The non-contiguity has been experimentally confirmed for three cases, eliminating the possibility that they are genome assembly artifacts. The formation of these complex chromosomes is currently under investigation and will be reported in detail separately. The

rearrangement events have been accounted for in the MIC/MAC comparative genome browser described above (http://www.jcvi.org/jbrowse/?data=tta2mic).

Nearly half the 225 Cbs's have the consensus C-rich strand sequence: 5'-TAAACCAACCTCTTT-3', and none has more than two substitutions to this sequence (*Table 2*). Confirming earlier studies (*Hamilton et al., 2006b*), 10 of the 15 nucleotide positions are completely conserved, while five show limited degeneracy, summarized as follows: 5'-WAAACCAACCYCNHW-3' (W = A or T; Y = C or T; H = A, C or T; N = any nucleotide; *Figure 3*). Cbs's identified in several related tetrahymenine species ([*Coyne and Yao, 1996*]) and below) fall within the same range of variability. All the positions occupied by T's in the consensus (found mostly toward the 3' end), and only these positions, exhibit some degeneracy. Only at positions 13 and 14 have we observed more than one type of substitution (13T→A, C, or G, 14T→A or C).

The limited Cbs degeneracy may reflect the specificity of the yet to be identified trans-acting factor(s) that physically interact with the Cbs. Pot2p is the first factor shown to associate specifically with Cbs regions in vivo, at the time of chromosome breakage (*Cranert et al., 2014*). Pot2p is a paralog of Pot1p, which is required for telomere maintenance. Pot2p may recruit factor(s) required for chromosome breakage and/or de novo telomere addition. As previously noted for the consensus sequence (*Yao et al., 1987*), every functional Cbs contains a permuted copy ($C_2A_2C_2$) of the *T. thermophila* telomeric repeat $C_4A_2$. More generally, the Cbs consensus shares with *Tetrahymena* telomeric repeats a striking C vs. G strand asymmetry; of the 117 non-consensus functional Cbs sequences, only one contains a substitution on the C-rich strand to a G (at position 13) whereas 27 contain a substitution to C (*Table 2*). The likelihood of this ratio being due to chance alone is low (probability of chi square << 0.01). Whether these sequence parallels between Cbs and telomeres are coincidental or related to Cbs function may be established when the mechanisms of chromosome breakage and telomere addition are better understood.

Many innovations in the realm of programmed genome rearrangement have resulted from the domestication of genes originally associated with mobile DNA elements; examples are found in

**Table 2.** Variation within the Cbs family. Pink and gray shading: single- and double-substituted variants, respectively.

| Cbs designation | Count | Cbs nucleotide position | | | | | Number of substitutions | Total substitutions per subset |
|---|---|---|---|---|---|---|---|---|
| | | 1 | 11 | 13 | 14 | 15 | | |
| canonical | 109 | | | | | | 0 | 0: 109 |
| 1A | 53 | A | | | | | 1 | |
| 11C | 8 | | C | | | | 1 | |
| 13A | 7 | | | A | | | 1 | |
| 13C | 2 | | | C | | | 1 | |
| 14A | 9 | | | | A | | 1 | |
| 14C | 4 | | | | C | | 1 | |
| 15A | 10 | | | | | A | 1 | 1: 93 |
| 1A,11C | 5 | A | C | | | | 2 | |
| 1A,13A | 2 | A | | A | | | 2 | |
| 1A,13C | 1 | A | | C | | | 2 | |
| 1A,14C | 2 | A | | | C | | 2 | |
| 1A,15A | 8 | A | | | | A | 2 | |
| 11C,13A | 1 | | C | A | | | 2 | |
| 11C,13G | 1 | | C | G | | | 2 | |
| 11C,14C | 1 | | C | | C | | 2 | |
| 11C,15A | 1 | | C | | | A | 2 | |
| 14A,15A | 1 | | | | A | A | 2 | 2: 23 |
| Total | 225 | 71 | 17 | 14 | 17 | 20 | | |

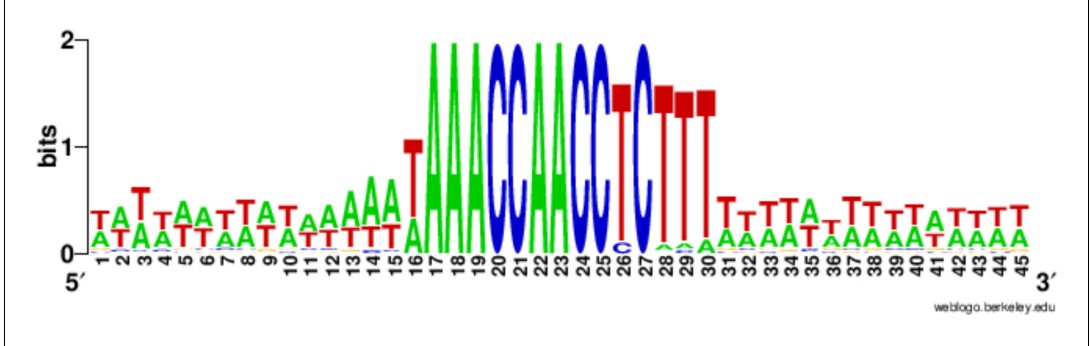

**Figure 3.** Conservation of the 15 bp chromosome breakage sequence. Nucleotide conservation was calculated at every position, as described in (*Hamilton et al., 2006a*), for the 225 Cbs's and their 15 bp flanking sequences, aligned on the C-rich Cbs strand. The Cbs element occupies positions 16 to 30. At any given position in the logo plot, two bits represent maximum conservation (only one nucleotide occupies that position), and 0 bits corresponds to no conservation (all four nucleotides are equally frequent).

multicellular organisms (*Kapitonov and Koonin, 2015*) and microbial eukaryotes (*Barsoum et al., 2010*; *Koufopanou and Burt, 2005*; *Levin and Moran, 2011*; *Sinzelle et al., 2009*), including ciliates (*Baudry et al., 2009*; *Cheng et al., 2010*; *Vogt et al., 2013*). The Cbs resembles the target site of a homing endonuclease, with its relatively long, non-palindromic sequence and limited degeneracy (*Fan and Yao, 2000*; *Hamilton et al., 2006b*); another superficial resemblance is to transposase binding sites found at transposon termini. It seems likely that Cbs and the yet unknown protein(s) that recognize it and initiate breakage had their origins in a mobile DNA element that invaded the germline genome and was subsequently domesticated.

## Conservation of chromosome breakage sites across *Tetrahymena* species

Cbs-mediated chromosome breakage has only been found in tetrahymenine ciliates. Earlier studies of this group (*Coyne and Yao, 1996*) showed strong evolutionary conservation of the Cbs sequence, but only one or two Cbs's per species were sequenced. To examine the evolutionary conservation of Cbs sequences and their locations within the germline genome, we conducted a pilot study of 12 consecutive breakage site locations in *T. thermophila* and three other *Tetrahymena* species, using the strategy described in Materials and methods (a more comprehensive study will be published separately). Strikingly, MAC chromosome ends were highly conserved in all four species, indicating strong conservation of breakage sites. Indeed, with just one exception in *T. borealis*, the location of every chromosome breakage site in the four species has remained identical since their divergence, down to the MIC genome interval between the same two consecutive homologous genes (*Supplementary file 2C*). The only detected differences are the deletion of DNA sequences surrounding *T. borealis* Cbs 3L-25 and a novel breakage site in *T. malaccensis*, between Cbs 3L-24 and 3L-25 (numbered according to *T. thermophila*). MAC chromosome lengths in this region are also strongly conserved among all four species (*Figure 4A*, *Supplementary file 2D*).

We sequenced the MIC Cbs regions for 22 of the 27 novel species/breakage-site combinations (see *Figure 4B*). No previously unidentified Cbs variants were observed in the 26 sequenced Cbs's (which include four locally duplicated Cbs's, see below). Importantly, there was consistency in the specific Cbs isoform found at a given breakage site in all four species, as expected if they represent a clade descended from a common ancestral Cbs at that site (see *Figure 4C*). This conclusion is further supported by the observation that Cbs's at a given homologous breakage site display the same orientation with respect to MAC-retained flanking regions, with the single exception of *T. borealis* Cbs 3L-22 (*Figure 4B and C*). In contrast to the conservation of the Cbs itself, there is little or no conservation of the 200 bp of adjacent sequence (not shown). Assuming the most parsimonious number of mutations to explain the Cbs variants observed at these nine homologous breakage sites, the rate of fixation of functional Cbs mutations is low; 11 mutations can account for all the Cbs variation observed at 31 independently sequenced sites (*Figure 4C*). This represents about 1.4 mutations

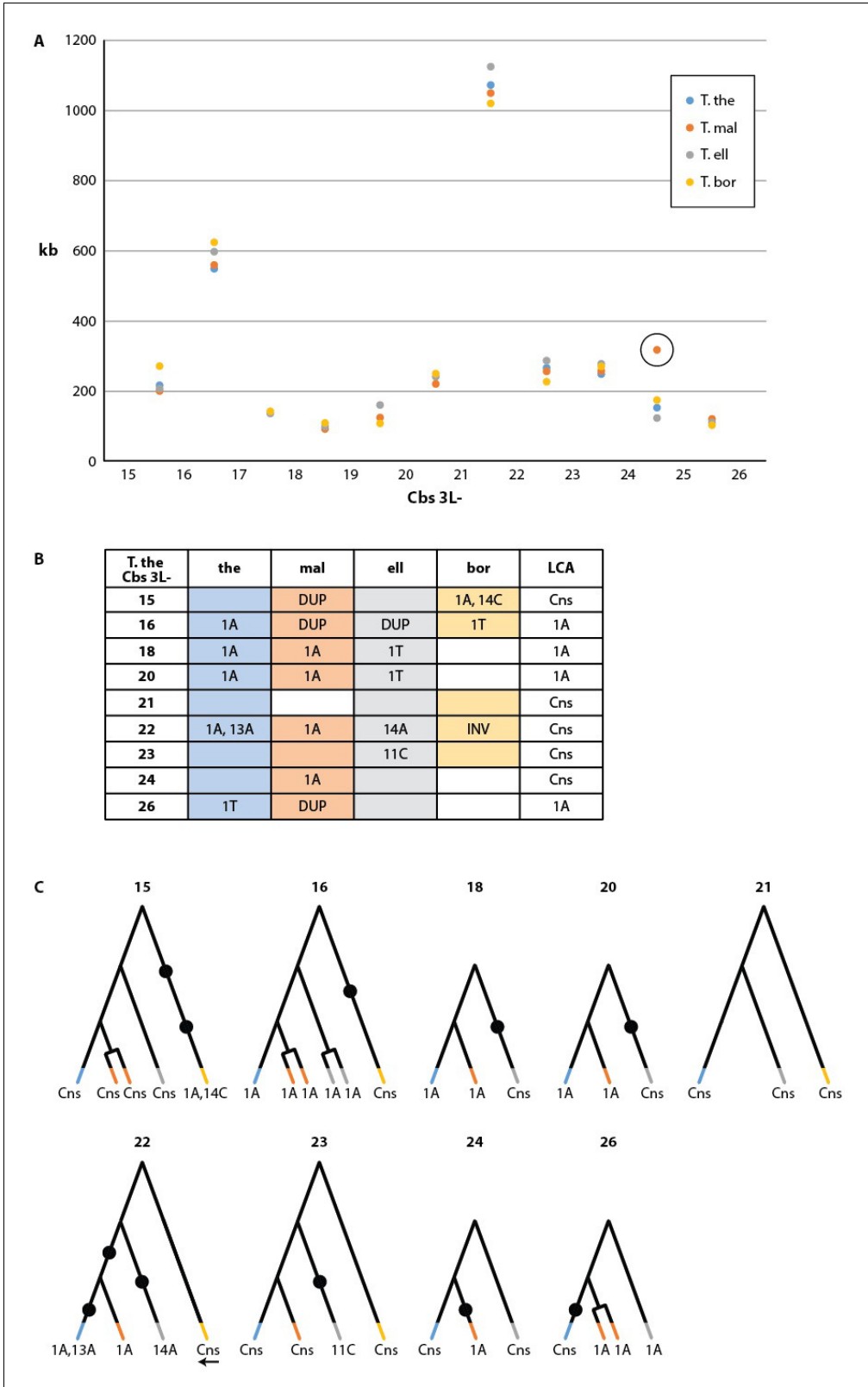

**Figure 4.** Conservation of chromosome breakage sites and Cbs in four *Tetrahymena* species. (**A**) Conservation of MAC chromosome lengths: X-axis: Cbs 3L-15 to 26 (evenly spaced). Y-axis: Length of the MAC scaffolds in each species whose ends are defined by the flanking Cbs's. Circle: an extra Cbs site in *T. malaccensis* creates two MAC chromosomes in this region; length = sum of the two MAC chromosome lengths. (**B**) Summary of Cbs sequence data at nine chromosome breakage sites; filled in box = sequence available; if no text = single, consensus Cbs in same orientation as *T. thermophila*; *Figure 4 continued on next page*

*Figure 4 continued*

Cbs sequence variants, duplications (DUP) and inversion (INV) indicated; final column = possible last common ancestor (LCA) Cbs, requiring a minimum number of mutations in the clade. (**C**) Inferred possible descent from Cbs of LCA at each of the nine chromosome breakage sites. Branch tips: Cbs consensus (Cns) or variant in *T.the*, *T.mal.*, *T.ell.*, and *T.bor*, in that order (colors consistent with parts A and B; missing branch = unsequenced Cbs). Terminally split branch = local Cbs duplication. Dots indicate minimal number of mutational events; placed in the longest branches when there is a choice. Reverse arrow (*T. bor.* 3L-22) indicates Cbs inversion.

fixed per breakage site since the divergence of these four species (corrected for eight unsequenced Cbs's and counting locally duplicated Cbs copies – see below – only once).

The retention of a functional Cbs sequence at each of the studied sites, in contrast to the divergence of immediately adjacent sequence, provides evidence that Cbs's, and therefore the positioning of chromosome breakage sites, are under purifying selection. In other words, it is functionally relevant to preserve the lengths and/or contents of specific MAC chromosomes. Further speculations on the possible nature of such selection are presented in the Appendix. The conservation of Cbs locations, and consequently the lengths of MAC chromosomes, is quite remarkable in contrast to the extremely variable locations of IESs, even in *T. malaccensis*, the species most closely related to *T. thermophila* (**Huvos, 2007**).

## Duplication of Cbs regions on an evolutionary time scale

Several reports document duplications of Cbs and surrounding sequences in *T. thermophila* (**Cassidy-Hanley et al., 2005**; **Hamilton et al., 2006b**; **Yao et al., 1987**). The pilot study described above revealed four Cbs duplications among the 22 sequenced sites in three other *Tetrahymena* species (**Figure 4B**). There are also reported cases of inter-species Cbs inversions (above and [**Coyne and Yao, 1996**]). Further analyses of such events may shed light on aspects of Cbs evolutionary history. We searched for Cbs-associated germline rearrangements genome-wide and identified a large number of both local, tandem repeat duplications and long-range duplications/translocations (**Supplementary file 2E**; summarized in **Table 3**). Forty-nine Cbs-containing segments (23% of the 225) have at least one duplicate in the MIC genome. Evidence for an earlier duplication/translocation of the entire rDNA locus, including flanking Cbs's, is described in the Appendix. This high frequency of Cbs-associated germline rearrangements supports the previous suggestion (**Coyne and Yao, 1996**) that some mechanism increases the likelihood of such events in the vicinity of Cbs, perhaps as a result of occasional missorting of the chromosome breakage machinery to the germline nucleus. Tandem repeat duplications appear concentrated within and at the margins of putative centromeres, perhaps reflecting an increased tendency of that chromatin domain to engage in such illegitimate recombination events, or a higher tolerance for their consequences.

Tandem duplications often generate predicted MAC chromosomes that are not maintained in the MAC (see next section). Long-range duplications result in widely separated duplicate Cbs pairs, on either the same or different MIC chromosomes. Such events would increase the maintained chromosome number (unless translocation occurs to the neighborhood of a pre-existing Cbs). As an example noted above, a novel breakage site in *T. borealis* was possibly introduced by long-range duplication. More complete analysis of MAC synteny among several *Tetrahymena* species, revealing genome-wide patterns of chromosome breakage conservation, will be presented separately. Future availability of MIC genome assemblies of these species will allow greater understanding of the frequency and consequences of long-range Cbs duplications.

The 49 Cbs-containing segment duplicates cluster into 15 sequence similarity groups (**Supplementary file 2E**; summarized in **Table 3**), which we call 'clades', to suggest that all members of each group were derived by successive duplications of an ancestral Cbs-containing segment. Some of these events probably occurred fairly recently, judging by the high-sequence identity of the Cbs-flanking regions. Within 14 out of 15 clades, members either have the same Cbs isoform (seven clades) or differ by a single substitution (seven clades). At least two mutations are required to explain the variation within the remaining clade. The doubly substituted 11C,13A Cbs is a relatively rare isoform found in more than one clade, suggesting they may form a 'super-clade'. Further observations on Cbs duplication, including evidence for at least one, and possibly two, super-clades, and a model of Cbs-mediated chromosome breakage evolution are presented in the Appendix. By back-

**Table 3.** Summary of salient features of *T. thermophila* Cbs clades.

| Cbs clade | Cbs Members [1] | Expect value range | Tandem duplications (repeat size) | Number of NMCs [2] | Inter-chromosomal duplications |
|---|---|---|---|---|---|
| 1L-1 | 1L-1 to 1L-5 | E-22 to E-48 [3] | 144 bp [4] | 4 | |
| 1L-16 | 1L-16, 4R-24 | E-47 | | 0 | 1L-4R [5] |
| 1L-17 | 1L-17, 1L-18, 1L-19, 4R-25 | E-18 to E-46 [3] | 45 bp | 2 [6] | 1L-4R [5] |
| 1L-20 [7] | 1L-20, 3L-14 | E-14 | | 0 | 1L-3L |
| 1L-28 | 1L-28, 1L-29 | E-18 | 530 bp | 1 | |
| 1R-1 | 1R-1 to 1R-7, 2L-2, XX-1, XX-3 | E-21 to E-178 | 13.6 Kb | At least 8 | 1R-2L |
| 1R-35 [8] | 1R-35, 1R-36 | E-139 | 469 bp | 1 | |
| 1R-37 [8] | 1R-37, 1R-38 | E-18 | 796 bp | 1 | |
| 2R-1 | 2R-1, 2R-2 | E-66 | 605 bp | 1 | |
| 3L-3 | 3L-3, 3L-29 | E-175 | | 0 | |
| 3L-4 | 3L-4, 4L-2, 4L-3 | E-48 to E-85 | 3.8 Kb | 1 next to 3L-4, 1 between 4L-2 and 4L-3 | 3L-4L |
| 4R-3 | 4R-3 to 4R-7, 4R-38, XX-2, XX-4 | E-49 to E-171 | 17.5 Kb | At least 6 | |
| 5L-9 | 5L-9, 5L-10, 5L-11 | E-58 to E-76 | 10.4 Kb | 2 | |
| 5R-5 | 5R-5, 5R-6 | E-22 [3] | 53 bp | 1 [6] | |
| 5R-14 [7] | 5R-14, 5R-15 | E-14 | 84 bp | 1 [6] | |

All clades are described in greater detail in **Supplementary file 2E**.

[1] Exact MIC supercontig locations of each Cbs are given in **Supplementary file 2A**.

[2] The number of predicted non-maintained chromosomes (NMCs) is one less than the number of repeat units.

[3] Shorter query length – expected values are potentially higher than for most alignments, which are based on a query length of 415 bp, for the same degree of sequence conservation.

[4] Average repeat unit length is artificially increased because an additional repeat unit containing a mutationally disabled Cbs between Cbs 1L-4 and Cbs 1L-5.

[5] Simultaneous duplication event; see **Supplementary file 2E**.

[6] These NMCs may be too short to be telomerized after chromosome breakage.

[7] Support for these clades is weaker than for the others; see **Supplementary file 2E**.

[8] This pair of adjacent clades may be a single clade.

**Source data 1.** MIC DNA sequences surrounding Cbs sites.

extrapolation, the identification of clades and possible superclades supports the suggestion that all current Cbs's are derived from one, or a few, founder copies present in a tetrahymenine ancestor. As described above, this founding event may have resulted from the invasion of the germline genome by a mobile element, followed by the domestication of an element-encoded gene to take over the mechanism of chromosome breakage from a pre-existing, less precise mechanism, such as that which persists in *Paramecium*.

## Non-maintained MAC chromosomes

Our previous studies (*Cassidy-Hanley et al., 2005*) identified two NMCs present in early sexual progeny (at 20 fissions after conjugation) but absent by ~120 fissions. A number of developmental events occur during this interval, including programmed genome rearrangement, the establishment of MAC chromosome copy number (*Doerder and DeBault, 1978*), and the transition from sexual immaturity to maturity (*Bleyman and Simon, 1967*; *Rogers and Karrer, 1985*). Because NMCs may play a role in these events, or serve as a model for their study, we undertook a genome-wide survey to identify more candidate NMCs and examine their properties. Using our MIC chromosome super-

assemblies, we identified a total of 33 NMCs (*Supplementary file 2F*), operationally defined as MIC DNA segments delimited by two consecutive Cbs's and absent from the MAC genome assembly.

To determine whether NMCs might contain genes that could function during conjugation or early post-conjugational development, we performed gene annotation on all NMCs greater than 1 kb in length. We identified 47 predicted genes, distributed among 10 NMCs (*Supplementary file 2G*). Some of them are homologous to genes found in the MIC-limited TEs REP and Tlr (*Fillingham et al., 2004*; *Wuitschick et al., 2002*), and others were annotated as transposases; therefore, the regions bearing these genes are likely recognized by the mechanism for IES removal and may, in fact, be processed as such, even though their flanking regions are not retained in the mature MAC. Nonetheless, RNA-seq evidence suggests that some NMC genes are expressed (see *Supplementary file 2G*) and may give rise to protein products that function during late conjugation and/or subsequent vegetative multiplication, until the NMCs are lost. For example, the five predicted genes with annotated transposase domains (one a piggyBac transposase, related to the domesticated transposase required for IES excision) may be involved in programmed somatic genome rearrangement (see below). The existence of expressed genes in transiently maintained NMCs may provide a novel mechanism for developmental gene regulation.

The mechanism(s) by which NMCs are lost from the MAC genome are of interest from the perspective of MAC chromosome maintenance, a poorly understood process involving DNA replication initiation and copy number control. We demonstrated that, in exconjugants at 24 hr post-mixing, all 20 NMCs longer than 1 kb have acquired telomeres (data not shown). Thus, telomere addition is not sufficient to fully stabilize these chromosomes. Moreover, whole genome sequencing data (not shown) from 24 hr exconjugants shows that all of the 13 largest NMCs (the only ones with sufficient read density for this determination) undergo developmental DNA endoduplication in concert with maintained MAC chromosomes, but it is currently unknown how rapidly NMC copy numbers decrease subsequently. The eventual loss of NMCs may result from the wide spacing of replication origins in the *Tetrahymena* genome. A recent study (*Gao et al., 2013*) identified roughly 7000 DNA segments that likely represent MAC DNA replication origins, an average of one per 15 kb, which corresponds to about 22.5 kb in the MIC genome, after adjusting for the average genome-wide IES fraction; only three NMCs are larger than this size. By contrast, in spirotrichous ciliates, a run-away evolutionary process, consistent with in silico predictions (*Morgens et al., 2013*), has led to extreme MAC chromosome fragmentation – down to gene-sized 'nanochromosomes'. This outcome was enabled by the evolution of independent DNA replication origins in association with nearly every gene in these ciliates. A lower origin density may have precluded such extreme chromosome fragmentation in *Tetrahymena*.

How do NMCs arise? Recent Cbs duplication appears to be intimately connected to the evolutionary origin of most currently observed NMCs; roughly 80% (26/33) of NMCs have Cbs's from the same clade on both sides (*Supplementary file 2F*). More than a third (13/33) of NMCs are short (1 kb or less) and contain no predicted genes; they likely have transient evolutionary existence, as their flanking, tandemly repeated Cbs's are functionally redundant and mutations are statistically almost certain to eventually inactivate one of the flanking Cbs's without penalty. However, it is possible that some very short NMCs could be maintained by selection if a defective Cbs near a newly broken end interfered with de novo telomere addition. Potentially, more interesting from an evolutionary perspective are the longer NMCs. A duplicated and translocated Cbs would have split a MAC chromosome into two fragments. The smaller one would become an NMC if it lacked cis-acting elements required for normal MAC chromosome maintenance. The resulting progeny would be viable if the NMC carried no genes essential for long-term vegetative multiplication. Over time, some of these developmentally short-lived MAC genes could undergo neo-functionalization for roles limited to post-zygotic and/or early post-conjugational development. Such NMC's likely would have greater longevity on an evolutionary time scale.

## Programmed DNA elimination

### Identification of IESs

Comparison of the *Tetrahymena* MIC genome assembly (157 Mb) to that of the MAC (103 Mb) indicates that about one third of the MIC genome is eliminated during MAC differentiation, considerably more than the 10–20% previously estimated by reassociation kinetic studies (*Yao and*

*Gorovsky, 1974*). We used three complementary methods to identify and map IESs, as described in 'Materials and methods' and *Figure 5—figure supplement 1*. We estimate the total number of IESs to be about 12,000, twice the estimate derived by extrapolation from a limited subset (*Yao et al., 1984*) or lower coverage MIC genome sequencing (*Fass et al., 2011*). The total DNA content within all identified IESs is around 46 Mb, accounting for 85% of the difference (54 Mb) between the MIC and MAC genome assemblies. This suggests that we have identified the majority of *Tetrahymena* IESs and that most MIC-limited regions are in the form of IESs.

As described in 'Materials and methods', the large sizes and repetitive nature of *Tetrahymena* IESs, along with inherent difficulties in assembling IES/MDS junctions, make it challenging to compile a list of IESs that is both comprehensive and precise in terms of deletion endpoints. To allow analyses of elements with precisely defined endpoints, we built a 'high confidence' set of 7551 IESs (*Supplementary file 3A*). These IESs correspond to 28.6 Mb of MIC DNA and range in length from 136 bp to 43.4 kb, with about 85% between 1 and 10 kb in length (*Figure 5—figure supplement 2*; mean = 3.78 kb; median = 2.78 kb). We rely on this high confidence set for all the following analyses

## Many IESs are related to transposable elements

Sequence similarity reveals little about the origins of many well-studied *Tetrahymena* IESs, but others show clear relatedness to TEs (*Chalker and Yao, 2011*). The MIC-limited regions of other ciliates also contain many TEs and TE-related sequences, supporting the hypothesis that programmed DNA elimination acts as a form of self-defense against genomic parasites (*Coyne et al., 2012*; *Klobutcher and Herrick, 1997*; *Vogt et al., 2013*). Thus, we analyzed our MIC genome assembly to determine the extent to which IESs are related to TEs and the nature of these relationships. This annotation revealed that putative TEs and their remnants make up approximately 18.6 Mb (12.6%) of the total MIC genome assembly, and 10.9 Mb (41.7%) of the high-confidence IES set (*Figure 6A*, *Figure 6—source data 1*, *2* and *3*, *Supplementary files 3B, 3C*). It is likely that an even higher proportion of IES sequences are ancestrally related to TEs, but have diverged too greatly for this relatedness to be detected using our criteria. Although 95% of putative TE sequences are removed through IES excision during the development of the new MAC, about 1 Mb of putative TE sequences appear to be retained in the MAC (*Figure 6A*; *Supplementary file 3C*). In some instances, we noticed a retention bias toward the terminal regions of the consensus of manually curated TE sequences (*Figure 6—figure supplement 1*). This suggests that sometimes the removal of TE sequences by IES excision is incomplete, leaving terminal sequences in the MAC.

Among all classified putative TEs, the vast majority corresponds to class 2 (DNA) transposons, which represent 48.7% of the repeated DNA in the MIC (*Figure 6B*; *Supplementary file 3C*). They belong to 'cut-and-paste' (3.6% of the MIC genome), *Helitron* (fragmented copies), and 'self-synthesizing' *Maverick*/Tlr (1.9% of the MIC genome) families (Supplementary Figure 6; *Supplementary file 3C*). Retrotransposons appear to be infrequent in *Tetrahymena*. Indeed, a small number (4.75% of the repeated DNA in the MIC) of non-LTR elements (mostly long interspersed nuclear elements, LINEs) was identified, but we found no evidence for the presence of any LTR retrotransposons. While retrotransposons are the predominant TE class in most eukaryotic genomes examined (mostly of plants, fungi, and animals; [*Huang et al., 2012*; *Levin and Moran, 2011*]), there are several other examples of eukaryotic genomes dominated by DNA TEs, including *Caenorhabditis elegans*, *Danio rerio*, and *Apis mellifera* (http://www.repeatmasker.org/genomicDatasets/RMGenomicDatasets.html).

Despite TEs being restricted to the transcriptionally silent MIC (*Chalker et al., 2013*; *Coyne et al., 2012*; *Schoeberl and Mochizuki, 2011*), we found evidence of very recent amplifications for the most abundant DNA cut-and-paste transposon superfamilies, based on the nearly identical sequences of some copies, as well as the presence of intact open-reading frames and terminal inverted repeats (see Materials and methods; *Figure 6C*; *Figure 6—figure supplement 2*). This suggests the recent transpositional activity of multiple DNA transposon families within the MIC genome.

## IESs show uneven spatial distribution, both locally and globally

As previously reported for individual IESs (*Austerberry et al., 1989*; *Li and Pearlman, 1996*; *Wells et al., 1994*) and more fully analyzed below, nearly all *Tetrahymena* IESs excise imprecisely.

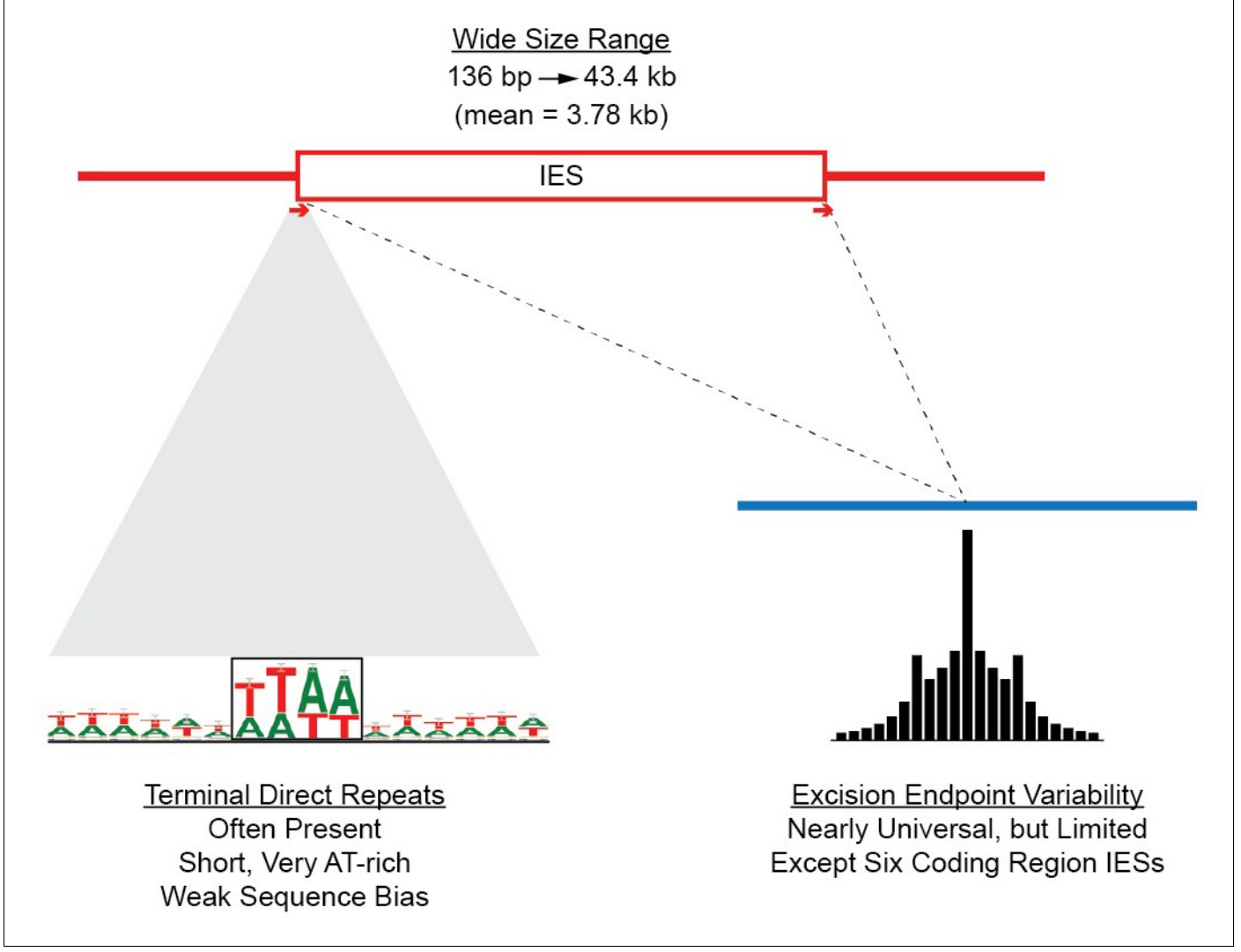

**Figure 5.** Summary of IES structural features. Red lines = MIC DNA. Blue lines = MAC DNA. A representative IES is indicated by the open red box. IESs were identified as described in *Figure 5—figure supplement 1*. Their size distribution is shown in *Figure 5—figure supplement 2*. The excision endpoint found in the SB210 MAC genome is indicated by the slanted lines converging to the right. Sequences from a large progeny pool representing multiple, independent excision events show most progeny share the parental endpoint, but variation within a limited range is common, as shown in detail in *Figure 5—figure supplement 3*. The left terminal junction sequences is shown blown up below and to the left. Short Terminal Direct Repeats (TDRs) are often found; they are generally very AT-rich and have a slight sequence pattern bias. A 4 bp TDR sequence logo is shown as an example. More detailed characterization of endpoint TDRs is presented in *Figure 5—figure supplement 4*.

The following figure supplements are available for figure 5:

**Figure supplement 1.** Read alignment methods used for IES dentification.

**Figure supplement 2.** Size distribution of 7551 high-confidence IESs.

**Figure supplement 3.** IES excision variability.

**Figure supplement 4.** IES/MDS junctions.

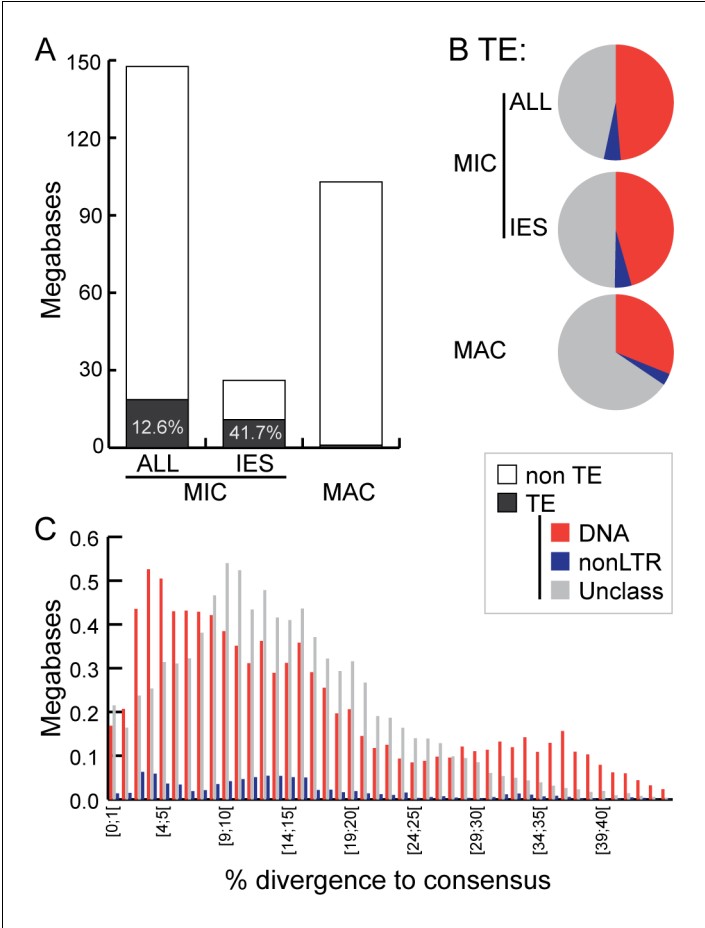

**Figure 6.** Transposable element landscape. (**A**) Proportion of DNA annotated as TEs (black) or unannotated (white) using RepeatMasker (**Smit et al., 2015**) and a custom putative TE library (see text). MAC putative TE content is about 1 Mb, potentially corresponding to a mixture of TE sequences retained in the MAC assembly and repeats not corresponding to TEs still in the library. (**B**) Proportion of putative TEs by class for MIC (ALL and high-confidence IESs) and MAC. In MIC(ALL), the most abundant elements (besides unclassified) correspond to DNA TEs ('cut-and-paste', *Mavericks* and Tlr elements). More than half of the MIC(ALL) non-LTR elements could be annotated as LINE1 elements. (**C**) Evolutionary view of putative TEs in the MIC. For each class, amounts of DNA are shown as a function of the percentage of divergence to the consensus (by bins of 1%), as a proxy for age: the older the TE invasion, the more copies will have accumulated mutations (higher percentage of divergence, right of the graph). Conversely, sequences corresponding to youngest elements show little divergence (left of the graph).

The following source data and figure supplements are available for figure 6:

**Source data 1.** Tetrahymena putative TE library.
**Source data 2.** Details of putative TEs contribution to the MIC chromosome super-assemblies.
**Source data 3.** Putative TE annotation of high-confidence 7551 IESs.
**Figure supplement 1.** MAC retention of TE termini.
**Figure supplement 2.** Landscape details of DNA TEs.

Therefore, unlike in some ciliates, *Tetrahymena* IESs are rarely found in MIC locations that give rise to MAC protein coding sequences; *Fass et al. (2011)* identified the only reported exceptions. To confirm these cases, search for others, and characterize IESs within introns, we first reannotated the protein-coding genes of the MAC genome. The improved gene models were then mapped onto the MIC genome sequence. As expected, virtually all the high-confidence IESs are removed from predicted intergenic (6182, 82%) and intronic (1168, 16%) regions, where imprecise excision would not cripple gene function. The remaining 2% mapped within putatively protein-coding gene sequences, but on closer inspection, most of these cases represent apparent annotation errors. We identified six solid cases of coding region IESs, described further below.

When the densities of IESs and putative TEs are plotted along the length of each MIC chromosome (*Figure 2*; green and blue lines, respectively), we observe a pronounced elevation in the central and terminal regions (accompanied by a corresponding depression in the density of predicted genes; *Figure 2* orange lines). We observe the reverse pattern on chromosome arms (although for unknown reasons, Chromosome 2 arms display higher IES density and correspondingly lower gene density than the other four). These results are consistent with the fact that shorter MIC scaffolds predominate at the middle and the ends of the MIC chromosomes, presumably because repetitive sequences in these regions make them difficult to assemble. The central regions, spanning approximately 7 to 12 Mb or about one quarter to one third of the chromosome lengths, share essentially the same range as the repeat-rich putative centromeric regions identified above. Thus, although *Tetrahymena* germline chromosomes are transcriptionally silent and carry dispersed elements destined for programmed elimination from the somatic genome, the general abundance of repetitive sequences and scarcity of genes in pericentromeric and sub-telomeric regions is similar to that observed in other eukaryotes (*Plohl et al., 2014*; *Pryde et al., 1997*).

The genomic distribution of IESs also bears on the regulation of programmed DNA elimination. IES excision in *Tetrahymena* relies on an RNAi-related mechanism, in which scnRNAs guide the epigenetic identification and targeting of IESs (*Mochizuki and Gorovsky, 2004*; *Vogt and Mochizuki, 2014*). We previously reported (*Noto et al., 2015*; *Schoeberl et al., 2012*) the existence of two distinct classes of scnRNAs, present at different stages of *Tetrahymena* conjugation. Early-scnRNAs recognize not just the IESs they stem from, but other IESs in trans. Early-scnRNAs also induce the production of Late-scnRNAs, and both types are cooperatively involved in DNA elimination (*Noto et al., 2015*). Early-scnRNAs are produced primarily from shorter MIC scaffolds, whereas Late-scnRNAs originate from both large and small MIC scaffolds, but the locations of these scaffolds on MIC chromosomes was previously unknown. We found that, whereas Late-scnRNAs originate from locations throughout the MIC chromosomes (*Figure 7B*), Early-scnRNA primarily map to the middle and end regions (*Figure 7A*). The increased number of IESs in these regions (*Figure 2*) does not fully explain this phenomenon, because many IESs are located within chromosomal arm regions, but most do not give rise to Early-scnRNAs. We conclude some yet unknown mechanism restricts production of Early-scnRNAs to the central and terminal chromosome regions.

In contrast with the wide and normally distributed size range of *Tetrahymena* IESs that we observe (*Figure 5—figure supplement 2*), *Paramecium* IESs in the MAC/MIC colinear portions of the genome are highly skewed toward shorter lengths (*Arnaiz et al., 2012*), and evidence suggests that progressive shortening occurs with age. Despite this shortening, there are relatively few documented cases of IES loss in *Paramecium* (*Arnaiz et al., 2012*; *Catania et al., 2013*). In contrast, in the few cases studied, *Tetrahymena* IES positions appear to be highly variable between species, even those most closely related (*Huvos, 1995*, *2007*). We hypothesize that *Tetrahymena* IESs proliferate in the MIC genome by TE movement (which our results suggest to be an ongoing process fueled by multiple invasions of diverse TEs), selectively constrained by the imprecise IES excision mechanism to intergenic and intronic positions. Unlike in *Paramecium*, there is not strong selection for a reduction in IES size. As *Tetrahymena* TE sequences gradually degenerate (and in the absence of both a precise excision mechanism and selective constraint on the precision of excision), the boundaries of IES removal shift, giving rise to the observed inter-species variability. Still, the small-RNA-mediated trans recognition network we have previously described and further refined above ensures that the overall pattern of IES excision is robust and reproducible from one sexual generation to the next. Because of the whole genome MIC/MAC comparison step in DNA elimination, this robustness is most likely necessary to avoid a complete breakdown in reproducibility, an event that might lead to reproductive isolation and speciation.

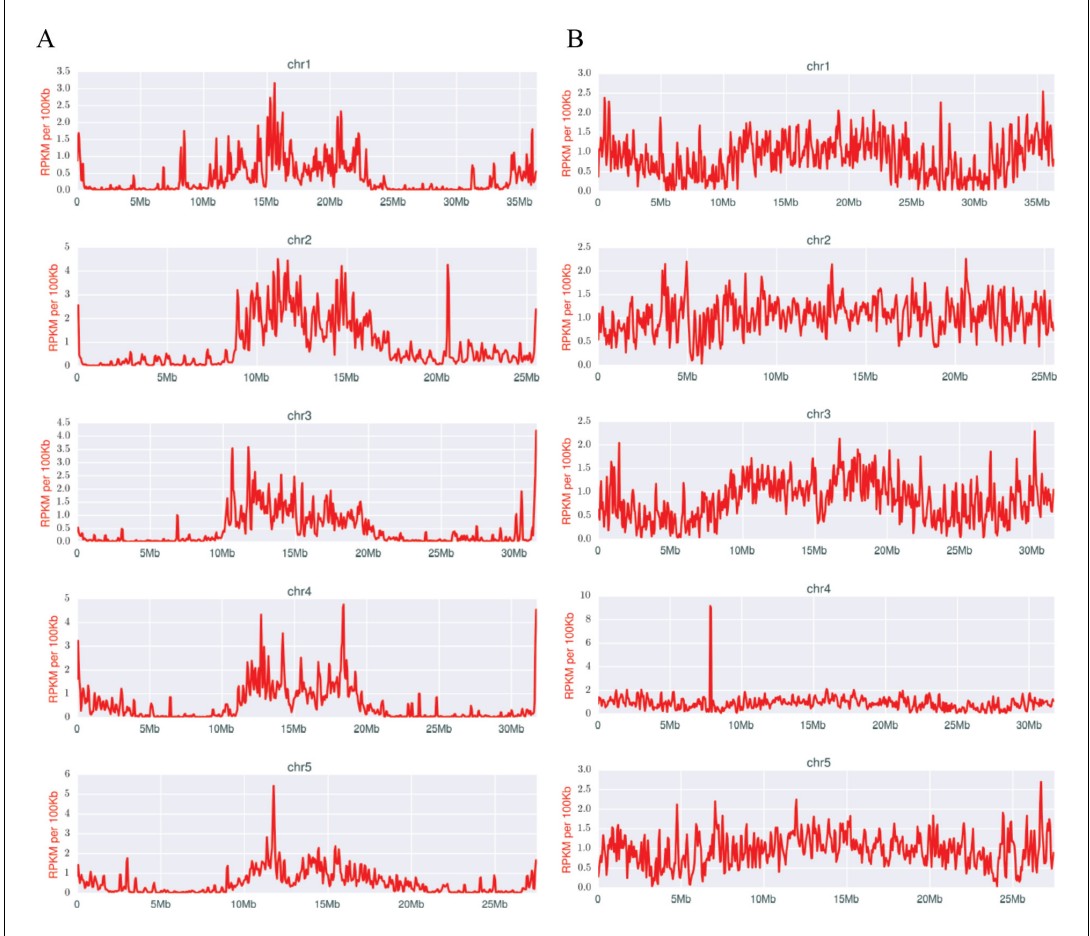

**Figure 7.** Densities of early (**A**) and late (**B**) scnRNAs on MIC chromosomes. X-axis = position on MIC chromosome super-assembly; all graphs normalized to the same length. Early-scnRNAs were co-purified with Twi1p at three hpm and Late-scnRNAs with Twi11p at 10.5 hpm. Normalized numbers (Reads per kb per million reads [RPKM] in 50 kb bins) of sequenced 26–32-nt RNAs that uniquely map to the MIC genome are shown. A few locations on the chromosomal arms where Early- or Late-scnRNAs were extensively mapped (e.g. ~20.6 Mb on Chr2 for Early-scnRNA and ~7.8 Mb on Chr4 for Late-scnRNAs) were examined in detail, but we have failed to detect any obvious unusual sequence features at these loci to account for the observed enrichment.

### New insights into *Tetrahymena* IES excision mechanism

Biochemical studies suggest that *Tetrahymena* IES excision occurs by a transposase-related mechanism initiated by a staggered, double-strand break (*Saveliev and Cox, 1996*, *2001*). Previous small-scale studies have shown that breakpoints do not share a strong consensus sequence and display frequent heterogeneity (*Austerberry et al., 1989*; *Li and Pearlman, 1996*). In some cases, IESs have short terminal direct repeats (TDRs) at their ends, one repeat remaining in the MAC following excision. The domesticated *piggyBac* transposase Tpb2p, required in vivo for IES excision, can introduce breaks in vitro of the expected geometry, and with a relaxed sequence preference (*Cheng et al., 2010*). To shed further light on this process, we examined thousands of additional IES junctions and conducted a genome-wide study of excision variability (*Figure 5*).

To investigate the range of variability in excision endpoints genome-wide, we purified and sequenced MAC DNA from a large pool of progeny from a mating between strains SB210 and SB1969, both belonging to the same inbred strain; MAC DNA from both parental strains served as controls. We aligned the sequencing reads to SB210 MIC scaffolds (as in *Figure 5—figure supplement 1*) to identify excision endpoints from multiple independent rearrangement events. Quantifying the degree of endpoint variation in a progeny pool depends on the experimental setup as well as the choice of validation criteria (as described in *Figure 5—figure supplement 3* legend), making

it hard to assign precise values. Nevertheless, even using conservative validation criteria, the great majority of IES sites exhibited variability, with up to 14 different junctions per site (*Figure 5—figure supplement 3A*). Even for IESs that at first appeared to have no, or exceptionally low, variability, closer visual inspection of alignment data revealed that nearly all exhibited some endpoint variability. The scnRNA-mediated genome rearrangement mechanism involves whole genome comparison of the parental and newly developing MACs, with the state of the pre-rearranged parental genome influencing events in the progeny in a locus-specific manner (*Mochizuki and Gorovsky, 2004*; *Yao and Chao, 2005*). Therefore, we examined the relationship of progeny to parental IES excision endpoints. Progeny endpoints were most often identical to those of the parent (26.6%), with most variations (83.3%) falling within 20 bp of the parental position (*Figure 5—figure supplement 3B*). This observation is consistent with the proposal that cis-acting 'boundary elements' act to prevent the spread of chromatin marks specific to MIC-limited sequences (*Chalker et al., 1999*; *Godiska et al., 1993*; *Li and Pearlman, 1996*; *Patil and Karrer, 2000*). The progeny endpoint distribution shows a small spike at a distance of 4 bp from the parental endpoint. This would be consistent with the use of the same breakpoint, followed by differential repair of the four base overhang generated by Tpb2p, using either the 'right' or 'left' overhang as template.

The greatest number of IES junction sites (28%) displayed no TDRs, and another 47% displayed TDRs of between 1 and 4 bp (*Figure 5—figure supplement 4A*). These TDRs are more AT-rich than immediately adjacent regions (*Figure 5—figure supplement 4B*), but include a wide diversity of sequences, with minimal bias (*Figure 5—figure supplement 4C*). *Paramecium* also initiates IES excision by the action of a domesticated piggyBac transposase thought to be monophyletic with Tpb2. However, in contrast to the junctional diversity observed in *Tetrahymena*, the TA dinucleotide central to the four nucleotide overhang is invariant (*Arnaiz et al., 2012*; *Gratias and Bétermier, 2003*) and excision endpoint variability is extremely low. These features have allowed *Paramecium* IESs to frequently occupy protein-coding regions, whereas the imprecision of Tpb2p and the near total absence of IESs in *Tetrahymena* protein-coding regions have most likely co-evolved to result in a strikingly different MIC genome landscape.

As mentioned above, we identified six Tetrahymena IESs that do fall within protein-coding regions, including three previously identified cases (*Fass et al., 2011*) (*Figure 8A*). These six IESs share four features that set them apart from the vast majority of other IESs. First, all six are flanked by TTAA terminal direct repeats (TDRs), one copy of which is retained in the MAC; this sequence feature is shared by only 2% of all *Tetrahymena* IESs. Second, these six IESs have a distinctive terminal inverted repeat (TIR), internal to the TTAA direct repeat, with a consensus of 5'- CACTTT-3' (*Figure 8B*, *Supplementary file 3D*). This TIR resembles that of *PiggyBac* TEs of several species (as found in RepBase: http://www.girinst.org/repbase/index.html and (*Xu et al., 2006*), and also the two full-length *piggyBac* consensus sequences annotated in the *Tetrahymena* MIC genome (*Figure 6—source datas 1*; 5'-CCCT(A/T)T-3' for Contig[0117] and 5'-CCC(A/T)(C/T)T-3' for R = 3481). Third, the six coding region IESs are all exceptionally short; in fact, they include the three shortest IESs we identified (136, 188 and 194 bp, *Supplementary file 3D*). Apart from their size and terminal sequences, no other conserved sequence features were detected, either within or flanking the IESs. Finally, these six IESs share the feature of exceptionally precise excision (as determined in the study of excision variability described above), as would be expected in order to maintain correct protein-coding capacity.

It has been reported that Tpb2p exhibits very little sequence specificity and wide variability (*Cheng et al., 2010*; *Vogt and Mochizuki, 2013*), so how does one explain the conserved junctions and precise excision of these few elements? We propose three testable hypotheses: first, this particular terminal sequence may allow these IESs to be processed by Tpb2p with unusually high fidelity. Second, an additional trans-acting factor may increase Tpb2p's specificity for these IES junctions. Finally, precise excision of these IESs may rely on one of the other p*iggyBac* transposase homologs identified in the *Tetrahymena* genome (*Cheng et al., 2010*), one of which we have shown resides on a NMC (see above) and is expressed during conjugation.

A remaining question is why, as long as a mechanism exists for precise excision of *Tetrahymena* coding region IESs, is their occurrence so exceedingly uncommon, especially in comparison with *Paramecium*, which employs a related piggyBac transposase mechanism for precise excision initiation, or conversely why has any precise excision at all persisted in *Tetrahymena*? It was suggested, in the case of the LIA2 coding region IES (*Fass et al., 2011*), that its excision may represent a novel form

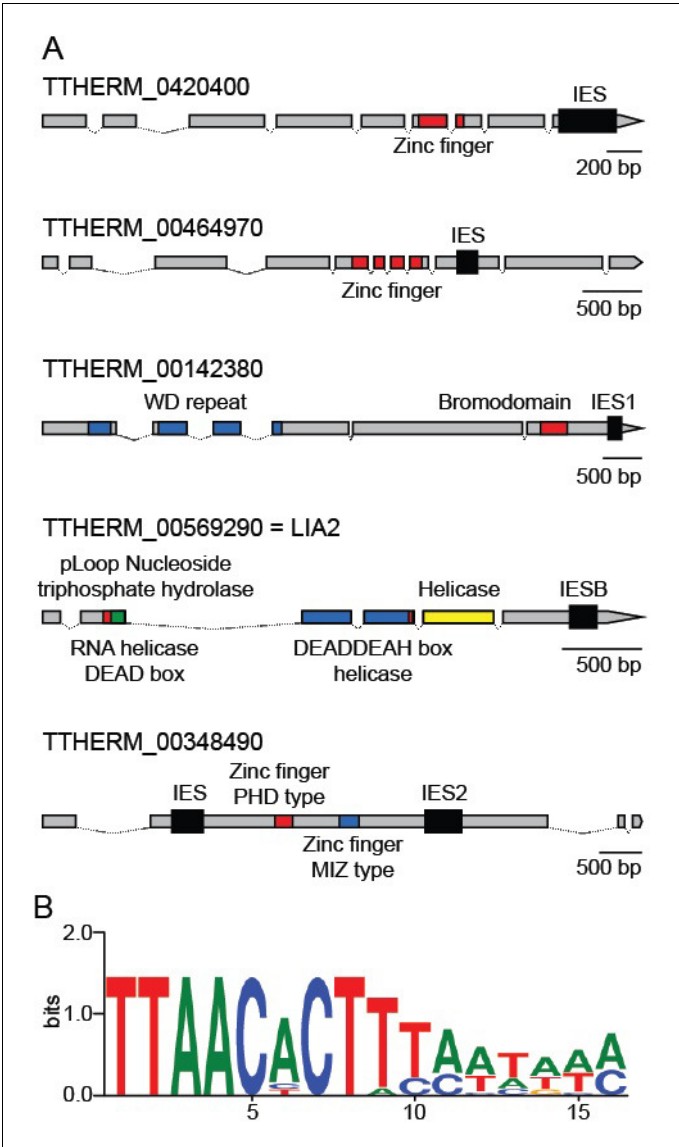

**Figure 8.** Coding region IESs. (**A**) MIC structures of the five genes containing coding region IESs (thick black boxes). Predicted protein-coding regions indicated by thinner boxes, conserved coding sequence domains by colored boxes, and introns by thin lines. Three coding region IESs previously identified (*Arnaiz et al., 2012*) are indicated as IESB, IES1, and IES2. (**B**) Sequence logo generated from the 12 IES/MDS junctions of the six IESs depicted in part **A** (interior of IES to the right). See also *Supplementary file 3D*.

of gene regulation, turning on or off expression or allowing the expression of different protein products before and after rearrangement. We note that the five genes containing coding region IESs share certain intriguing features, suggesting that such regulation could serve relevant function(s) (*Figure 8A*). The expression of all five is conjugation-specific (*Miao et al., 2009*; *Xiong et al., 2012*), peaking at approximately the same time point (8–10 hr post-mixing; http://tfgd.ihb.ac.cn/), near the time of IES excision. In addition, the predicted protein sequences contain functional domains of potential relevance to the regulation of gene function and/or MAC development, including zinc finger, bromodomain, and DEAD-DEAH helicase (Lia2p). Perhaps, the emergence of an efficient yet imprecise excision mechanism in *Tetrahymena*, based primarily on the epigenetic scnRNA mechanism to identify IESs and determine their boundaries, has driven selection against almost all coding region IESs that existed in a common ancestor of *Tetrahymena* and *Paramecium*, with the few

remaining *Tetrahymena* cases persisting for gene regulatory purposes. Future studies of these five genes and six IESs will test this hypothesis.

## Conclusions and future directions

The assembly and analysis of the germline genome of *Tetrahymena thermophila* provides many new insights into the architectural differences between this genome and the remodeled somatic genome, the evolutionary history that shaped these genomes, and the developmental rearrangement mechanisms through which the somatic genome matures. Consistent with the deep evolutionary branching of the phylum (*Baroin-Tourancheau et al., 1992*), comparisons between *Tetrahymena*, *Paramecium*, and *Oxytricha* reveals the extraordinary diversity and adaptability of ciliate germline/soma nuclear differentiation. Further comparative germline genomic analyses, including additional species closely related to the model organisms *T. thermophila*, *P. tetraurelia*, and *O. trifallax*, will help elucidate further details of this remarkable process. Because genome-wide DNA elimination occurs in phylogenetically diverse eukaryotes and is mechanistically related to nearly universal chromosomal functions mediated by small RNAs, the implications of this area of research are expected to be wide-ranging. The genome sequence, super-assemblies, and analyses presented here provide valuable information and resources for future investigations of programmed genome rearrangement and the relationship between chromosome structure and function in germline and somatic nuclei.

## Materials and methods

### Genomic library construction and sequencing

*T. thermophila* strains undergo occasional chromosome loss in their silent germline nucleus. To confirm that the sequenced SB210 strain (RRID:TSC_SD01539) isolate had all five micronuclear chromosomes intact, whole cell genomic DNA was isolated and germline-specific sequences on both arms of each chromosome were mapped (*Cassidy-Hanley et al., 1994*). From such a validated isolate (available from the *Tetrahymena* stock center; https://tetrahymena.vet.cornell.edu), micronuclei were purified and genomic DNA prepared according to published procedures (*Gorovsky et al., 1975*). By microscopic counting of purified nuclei (taking into account the relative nuclear ploidy), we estimate that contamination with macronuclear genomic DNA was less than 2%.

The *T. thermophila* germline genome is highly AT-rich (MAC assembly 77.7% and MIC assembly 77.9%) and contains abundant repetitive sequence elements, two factors that can complicate genome sequencing and assembly. To avoid PCR-based bias in sequence representation, we generated Illumina fragment sequencing libraries using a PCR-free protocol. Two Illumina whole genome shotgun PCR-free fragment libraries were generated following published procedures, using the 'with-bead' approach (*Fisher et al., 2011*; *Kozarewa et al., 2009*). Five microgram of genomic DNA was sheared to 150–300 bp using a Covaris LE220 instrument with the following parameters: temperature: 7–9°C; duty cycle: 20%; intensity: 5; cycles per burst: 200; time: 90 s; shearing tubes: Crimp-Cap microtubes with AFA fibers (Covaris Inc., Woburn, MA). Following DNA fragment end-repair and A-tailing, fragments were ligated on both ends with PCR-free-enabled TruSeq adapters (Illumina FC-121–2001) following manufacturer's recommendations (Illumina Inc., San Diego, CA). No PCR amplification was performed and resulting libraries were size-selected to contain inserts of 180 bp ± 10% with a Sage Pippin Prep using a 2% cassette following manufacturer's recommendations (Sage Science, Beverly, MA).

To maximize scaffold lengths, we added to the fragment reads an approximately equal number of mate-paired reads from six 'jumping' libraries. Six mate-pair jumping libraries were generated using Illumina's Mate Pair Library Preparation Kit v1 following the manufacturer's recommendations with the following modifications. Twenty microgram of genomic DNA was sheared to approximately 2–10 kb in size using a HydroShear (Digilab, Marlborough, MA) with the following conditions: cycles: 22; speed: 16; assembly: 0.002"; total volume: 200 µl 1x low TE buffer (10 mM Tris pH 8.0, 0.1 mM EDTA). Following end repair and biotin labeling, DNA fragments were separated on a 0.6% agarose gel and size fractions collected in the following approximate ranges (number of resulting libraries in parentheses): 2–4 kb (2), 4–5 kb (2), 5–6 kb (1), and 7–9 kb (1). Each fraction was processed individually, using indexed adapters, rather than standard paired-end Illumina sequencing adapters, to enable library pooling during sequencing.

The two PCR-free fragment and six jumping libraries were sequenced with 101 base paired-end reads using an Illumina HiSeq2000 instrument following manufacturer's recommendations. Sequencing generated a total of approximately 125 Gb of data.

## De novo genome assembly

The MIC genome was assembled with 169-fold sequence coverage using roughly an equal mix of fragment and mate-pair read data. Assemblies were generated with the ALLPATHS-LG assembler (*Gnerre et al., 2011*) (RRID:SCR_010742; version 38019) using default parameters. Assemblies were screened to remove single contig scaffolds smaller than 1 kb and contigs less than 200 bp in length. Assemblies were screened against the complete *T. thermophila* mitochondrial genome sequence using nucmer (from the Mummer package v3.23 64 bit package run with default parameters; RRID: SCR_001200) to identify and remove any mitochondrial contigs. MIC telomeres, which have a distinctive terminal repeat sequence as well as sub-telomeric repeats (*Kirk and Blackburn, 1995*), were not detected in the MIC genome scaffolds, suggesting that their repetitive nature prevented assembly.

## RNA-Seq library construction and sequencing

Total RNA was prepared from three *Tetrahymena thermophila* cell populations: strain CU428 (RRID: TSC_SD00178) in mid-log phase growth, the same strain in starvation medium (10 mM Tris-HCl, pH 7.4) at time t = 0 hr, and a mixture of conjugating pairs of CU427 (RRID:TSC_SD00715) and CU428 at times t = 3, 6, and 9 hr post-mixing. In each case, $5 \times 10^6$ cells were resuspended in 600 µl Trizol Reagent (Life Technologies-Thermo Fisher Scientific, Waltham, MA) and processed according to manufacturer's recommendations. Precipitates were resuspended in nuclease-free water and treated with TURBO DNase (Ambion-Thermo Fisher Scientific, Waltham, MA). Samples were then ethanol precipitated and resuspended in nuclease-free water and pooled.

An Illumina (Illumina, Inc., San Diego CA) RNA-seq library was prepared using the dUTP second-strand method (*Levin et al., 2010*) with the following modifications. Twelve microgram of total RNA was subjected to poly(A)$^+$ isolation using two rounds of purification with the Dynabeads mRNA purification kit (Invitrogen-Thermo Fisher Scientific, Waltham, MA). Poly(A)+ RNA was treated with Turbo DNase (Ambion) according to the manufacturer's recommendations and shown to be free of residual, detectable genomic DNA based on a qPCR assay (data not shown). The resulting 135 ng of poly (A)$^+$ RNA was then fragmented in 1x RNA fragmentation buffer (New England Biolabs, Ipswich, MA) at 85°C for 4 min. Following first strand cDNA synthesis, cDNA was purified with 1.8x RNAClean SPRI beads following manufacturer's recommendations (Beckman Coulter Genomics, Danvers, MA). Index Illumina sequencing adapters were used in place of standard paired-end adapters to enable library pooling during sequencing. Following adapter ligation, smaller library fragments were removed with two 0.7x AMPure XP SPRI bead purifications following manufacturer's recommendations (Beckman Coulter Genomics). PCR amplification was performed with Phusion High-Fidelity PCR Master Mix with GC Buffer (New England Biolabs) and 2 M betaine using the following cycling conditions: 30 s at 98°C; 9 cycles of 98°C for 10 s, 65°C for 30 s, and 72°C for 30 s; 5 min at 72°C. RNA-Seq libraries were sequenced with 101 base paired-end reads using an Illumina HiSeq2000 following the manufacturer's recommendations (Illumina). Sequencing generated a total of approximately 150M paired-end reads.

## Joining MIC scaffolds into chromosome-length super-assemblies by tiling method

All MAC and MIC scaffolds were aligned to one another using nucmer (criteria: percent identity >95, alignment length >1000 bp; Mummer package RRID:SCR_001200) and blastn (criteria: percent identity ≥98, alignment length ≥100 bp; RRID:SCR_001598) to identify regions of common origin between the two genome assemblies. Contiguous blocks of alignment (interrupted by IESs) to single MAC scaffolds were used to place MIC scaffolds in their natural order and orientation. To extend these contiguous blocks and join adjacent MIC or MAC scaffolds, we used a 'tiling' method, illustrated and described in *Figure 1—figure supplement 1*. We constructed 'best approximation' super-assemblies of all five MIC chromosomes by combining the scaffold alignment overlap data with HAPPY mapping results. We also incorporated findings on seven cases of programmed DNA rearrangement events that join non-contiguous MIC genome regions into MAC chromosomes (see

above). We found evidence suggesting a number of cases of MIC scaffold mis-assembly (*Supplementary file 1D*), but only two cases of MAC scaffold mis-assembly. This difference likely results from three factors: (1) the MIC genome contains many more repetitive sequences, (2) the MAC genome was assembled using long-read Sanger technology (*Eisen et al., 2006*), and (3) the MAC assembly underwent extensive finishing (*Coyne et al., 2008*). Scaffolds with ambiguous alignment placement patterns, due to repetitive sequences, were omitted from the resulting chromosome super-assemblies. When overlapping alignments were not available to bridge an intra-scaffold gap, HAPPY mapping data [(*Hamilton et al., 2006a*) and *Supplementary file 1B*] were used to place adjacent scaffolds in the best possible order.

The super-assemblies are summarized in *Supplementary file 1C* and are also available in a JBrowse format at: http://www.jcvi.org/jbrowse/?data=tta2mic (RRID:SCR_001004) and fasta format at: http://datacommons.cyverse.org/browse/iplant/home/rcoyne/public/tetrahymena/MIC. The five chromosome super-assemblies incorporate 765 of the 1464 total MIC scaffolds, but because most of the unincorporated scaffolds are small (83% < 10 kb; *Supplementary file 1C*), the super-assemblies account for 152 Mb of the 157 Mb total MIC assembly length. Over 60% of the unincorporated scaffolds have no significant matches to the MAC assembly. The rest have only very short (<200 bp) matches, suggestive of repetitive sequences, or else can be incorporated within larger MIC scaffolds, suggesting mis-assembly. Thus, it appears all or nearly all the unincorporated scaffolds are entirely MIC-specific.

## Identifying Cbs's

Previously identified Cbs's (*Hamilton et al., 2005*, *2006*) were confirmed and searches for additional Cbs's were done by two independent methods. First, we searched for Cbs family members in the MIC neighborhood that aligns with MAC chromosome telomere-addition sites. Second, we searched for Cbs's directly in MIC supercontigs using a Perl script and the following regular expression for the Cbs family: 'WAAACCAACCYCNHW', where W = A or T, Y = C or T, N = any nucleotide and H = A, C or T (*Hamilton et al., 2006a*).

## Detecting and clustering duplications of Cbs's and their adjacent sequence

To detect duplications of Cbs-containing regions, we used 415 bp DNA segments containing each Cbs at the center and 200 bp of adjacent sequence on each side, referred to as 'Cbs segments' (*Table 3—source data 1*). In four cases, the segments contained two or three Cbs's, as adjacent Cbs were less than 400 bp apart. In such cases, we used a region that contained all such Cbs plus 200 bp flanking the outer Cbs's. We used two independent methods to align and cluster Cbs segments. In method 1, the 225 segments were aligned with one another in all pairwise combinations using the 'Align two sequences' option of NCBI Blast (http://blast.ncbi.nlm.nih.gov/Blast.cgi; RRID:SCR_004870). Default parameters were used with the following exceptions: word size = 7, low complexity filter = OFF, and expect value threshold = 1E-15 Word size was set to seven because every functional Cbs has nine consecutive, absolutely conserved nucleotides. This choice ensured catching every alignment, and thus all segmental duplications that included the Cbs. The low complexity filter was set to OFF because non-genic portions of the *T. thermophila* genome are highly A+T-biased (the entire set of Cbs-flanking 400 bp segments is 86.6% A+T). Setting the low complexity filter to ON generally reduced the alignment to a small number of nucleotides on either side of the Cbs sequence and increased by one order of magnitude the exponent of the expected value of the best alignments. The 1E-15 threshold was empirically chosen by comparing expect values for alignments of the set of 225 Cbs segments with each other and with sets of randomized sequences; using this expect value threshold reduced spurious alignments between real and randomized sequences to a modest number. Randomized sequences were obtained as follows. For each Cbs segment, the Cbs was kept intact, but the 200 bp of flanking sequence on each side were randomized, while retaining the nucleotide frequency of the original sequence. Three independent randomized sets were generated. The center Cbs was kept intact to make the simulation equivalent to the real analysis. This was necessary because inclusion of a 15 bp highly conserved Cbs sequence results in a spurious decrease in the expect value of the alignment between any pair of Cbs segments.

The entire set of 225 sequences was used as the 'subject' database for all-by-all alignment. Smaller groups of the same sequences were used as 'query'. We confirmed that the expect value of the alignment of a given Cbs region was not affected by how many Cbs segments were simultaneously used as query. Since we were interested in duplications that include the Cbs, alignments were considered only if Cbs's aligned with one another. In a minority of cases (involving 11 Cbs's), consecutive Cbs are less than 400 bp apart. Because of sequence overlaps in the 415 bp segments centered on such closely-spaced Cbs's, the resulting self-matches would spuriously inflate the statistical significance of certain alignments and could lead to meaningless clusters. To deal with this issue, the individual Cbs-containing repeat units of closely-spaced-Cbs segments were split into non-overlapping pieces based on preliminary alignments and aligned with the 225 Cbs segments. As a consequence, some of the Cbs segments were significantly shorter than 415 bp, which raises the minimum expect values of their alignments. Cbs segments that aligned with at least one other segment with expect value equal or less than 1E-18 were clustered. We chose this threshold to exclude most false positives, based on the distribution of expect values for all-by-all alignments of the real sequences to the corresponding distribution for randomized vs. real sequences.

Method two started with an all-by-all Blastn alignment of the 225 Cbs segments also with low complexity filter = OFF. The expect value threshold was set at E-07. This threshold was obtained empirically by inspecting the clusters and alignments as described below, with a goal of having the largest clusters that still were conservative enough to generate good multiple sequence alignment. The Blastn matches were clustered using the mcl algorithm (http://micans.org/mcl/index.html?sec_thesisetc) (*Van Dongen, 2000*) with the pairwise scores set to the ratio of the bit score divided by the bit score of the sequence matched against itself (as is done in the IMG system [*Markowitz et al., 2006*]). The sequences in each cluster were then aligned using MUSCLE (*Edgar, 2004*) (RRID:SCR_011812).

Midpoint-rooted maximum likelihood phylogenetic trees for each cluster were generated using PhyML version 20120412 (*Guindon and Gascuel, 2003*) (RRID:SCR_014629). The GTR (*Tavaré, 1986*) nucleotide evolutionary model was used with a 4-category discrete gamma model of rates across sites. One hundred parametric bootstrap replicates were performed for each tree to measure robustness of the topology.

## Identifying Cbs's in other species

*T. thermophila* strain SB210, *T. malaccensis* strain 23b (RRID:TSC_SD01730), *T. elliotti* strain 4EA (RRID:TSC_SD01607), *T. borealis* strain X4H2 (RRID:TSC_SD01609; all strains available at the Tetrahymena Stock Center; https://tetrahymena.vet.cornell.edu) were grown in 2% PPYS. Whole-cell DNA was prepared by proteinase K digestion, phenol-chloroform extraction, and ethanol precipitation (*Bannon et al., 1983*). RNaseA treatment was either included with the proteinase K treatment or performed separately on purified DNA. Primers used for PCR amplification were obtained from Integrated DNA Technologies, Inc. (Coralville, IA). DNA sequencing was done by MCLAB (South San Francisco, CA).

The following strategy was used to identify Cbs's in *T. malaccensis*, *T. elliotti*, and *T. borealis* (for discussion of the relatedness of these species, see *Chantangsi et al. [2007]*). First, we chose a well assembled 3.4 Mb *T. thermophila* germline DNA segment, composed of contiguous MIC supercontigs 2.6 and 2.11. All but one of the 11 MAC chromosomes derived from this region initially assembled from telomere to telomere, and therefore, the region is unlikely to contain many repetitive sequences. This region is located on MIC chromosome 3 and contains 12 consecutive Cbs (3L-26 to 3L-15). To locate the chromosome breakage sites and identify the *T. thermophila* MAC chromosome scaffolds derived from this region, the MAC scaffold database at the NCBI *T. thermophila* Nucleotide Blast page (http://blast.ncbi.nlm.nih.gov/Blast.cgi?PAGE_TYPE=BlastSearch&PROG_DEF=blastn&BLAST_PROG_DEF=megaBlast&BLAST_SPEC=OGP__5911__12563) was searched with the two MIC supercontigs. Homologous MAC supercontigs from *T. malaccensis*, *T. elliotti*, and *T. borealis* were identified by using blastn (RRID:SCR_001598) to search their genome assemblies (Accession numbers: *T. malaccensis* (PRJNA51577), *T. elliotti* (PRJNA51573), and *T. borealis* (PRJNA51575) using 20–25 kb segments from the ends of each *T. thermophila* MAC chromosome identified above. Next, guided by the *T. thermophila* MIC chromosome 3 super-assembly, we constructed pseudo-assemblies of the homologous MIC chromosome segments for each of the other three species (called pseudoassemblies because they lack IES's, whose absence does not affect the connectivity of

the MAC-destined sequences). Based on the assumption that these pseudoassemblies are colinear with the *T. thermophila* super-assemblies, we predicted the location of MIC Cbs. A more complete analysis of synteny between these and other *Tetrahymena* species will be presented in a later paper.

To establish whether Cbs's are indeed present at the predicted sites, we followed the general Cbs sequencing strategy of *Hamilton et al. (2005)*. The MIC-specific region, expected to contain a Cbs, was PCR amplified from whole cell DNA using primers from the ends of each pair of MAC chromosomes predicted to be adjacent in the MIC. The amplified product was sequenced directly (without cloning), either after reaction cleanup or, if necessary, gel purification (Qiagen, Valencia, CA). With one exception, PCR products were sequenced from both strands or from reactions with different primer combinations. The sequence was then aligned with a short concatenated sequence from neighboring MAC chromosomes (telomeres or Ns deleted) to verify that the correct product was obtained and to delineate the MIC-specific segment. For alignments, we used LALIGN (http://www. ch.embnet.org/software/LALIGN_form.html; RRID:SCR_011819) and Needle (http://mobyle.pasteur. fr/cgi-bin/portal.py#forms::needle; RRID:SCR_008493), most often with default settings, although in some cases the gap start penalty was decreased. A Cbs, presumably functional (*Hamilton et al., 2006b*), was found in the middle of the MIC-specific segment of every genuine PCR product.

*T. elliotti* supercontigs 14, 30, and 46 presented special problems in the assembly; these supercontigs appear to be concatenates of homologs of several *T. thermophila* MAC chromosomes linked by blocks of N's at the sites where MAC chromosome ends are predicted. To determine whether these long supercontigs truly represent a single chromosome or were due to misassembly problems, we amplified regions of presumed fusion events with primers on both sides of the Ns. Amplification products were sequenced at locations homologous to 3L-26, 3L-21, 3L-20, and 3L-16; all four were found to contain known Cbs sequences.

To prove that the Cbs sequences are functional, we set out to find telomeres at the ends of MAC chromosomes that flank each Cbs identified in the three relatives of *T. thermophila*. This was done in one of the following ways: (1) Many of the MAC supercontigs had assembled telomere sequence (CCCCAA or its reverse complement) at one or both ends; (2) At ends where telomere sequence repeats did not assemble, or adjacent to blocks of Ns in *T. elliotti*, we were sometimes able to identify telomere repeat-containing sequence reads from the Sequence Read Archive (www.ncbi.nlm.nih. gov/sra) that aligned with the incomplete MAC supercontig assembly; (3) In other cases, a telomere was experimentally identified by PCR amplification using a primer specific to the terminal DNA sequence paired with a generic telomere sequence primer, as described by Hamilton et al. (*Hamilton et al., 2005*). The presence of a single (or major) PCR product, consistent with the predicted product size, was taken as evidence for the presence of telomeres at that chromosome end. For confirmation, some of these telomere-containing PCR products were cloned into the PCR2.1 vector using the Invitrogen TA cloning kit (Thermo Fisher Scientific, Waltham, MA) and sequenced. Several clones were sequenced in each case —as in *T. thermophila*, microheterogeneity in the telomere addition site was observed (*Fan and Yao, 1996*; *Hamilton et al., 2006b*); (4) A supercontig containing two MAC chromosomes separated by a block of N's was revealed in the case of the *T. elliotti* homolog of Cbs 3L-26; two bands were seen after hybridization to Southern blots of pulsed-field gels of *T. elliotti* DNA (data not shown). Using the same method, we indirectly confirmed the existence of Cbs 3L-21 by confirming the length of the MAC chromosome that flanked the Cbs. Using these methods, a telomere-capped MAC end was identified on at least one side of every Cbs studied, confirming Cbs functionality.

In order to verify at higher resolution the location of putative homologous Cbs's in the four species, we looked for homology of predicted protein-coding sequences flanking every one of the 12 identified or predicted Cbs's in the three species to the corresponding gene models in *T. thermophila*. This was done by aligning 2500–5000 bp of DNA sequence adjacent to each end of every MAC supercontig in the other species to the *T. thermophila* protein database at TGD using blastx (RRID:SCR_001653). This was done to circumvent any potential gene annotation errors in the three other species. For each of the 72 cases, we recorded whether or not the closest matching *T. thermophila* gene(s) were also the terminal Tel-adjacent gene(s) of the corresponding *T. thermophila* scaffold.

## Identification of tandem repeats

The location and characterization of tandem repeats were done using Tandem Repeats Finder (*Benson, 1999*) with default parameters, as follow: Match = 2, Mismatch = 7, Delta = 7, PM = 80, PI = 10, Minscore = 50, MaxPeriod = 500 (i.e. tandem repeats up to 500 nt can be found), with the option –h to obtain text file outputs. Tandem repeats of only two copies were not considered. To obtain an accurate amount of DNA covered by the tandem repeats, we used bedtools merge (*Quinlan and Hall, 2010*) (RRID: SCR_006646). By these criteria, we identified 29,794 simple repeats (constituting 2.9 Mb, or 1.9% of the MIC scaffold sequence, excluding Ns). However, this number is likely an underestimate of the overall MIC genome repeat content. Because of the numerous gaps in the MIC genome assembly, approximately 6.4% (constituting 10.0 Mb) consists of Ns. Because repeat-rich sequences are difficult to assemble, the unassembled parts of the genome are likely to contain many more repeats. By comparison, the MAC genome assembly, which is much more complete, contains far fewer Ns. Using the same Tandem Repeat Finder parameters as for the MIC, we identified 22,216 simple sequence repeats (constituting 3.5 Mb, or 3.4% of the MAC assembly, excluding Ns). Although the number of repeats is lower in the MAC than the MIC, there were many longer repeats. Thus, as first described by *Eisen et al. (2006)*, there is a significant retention of repetitive sequences in the MAC, but it is not possible to say what portion of these sequences are eliminated from the MIC genome.

## Identification and confirmation of IESs

As stated in the main text, the generally large size and repetitive nature of *Tetrahymena* IESs, along with inherent difficulties in assembling IES/MDS junctions (described below) make it challenging to compile a list of IESs that is both comprehensive and precise in terms of IES endpoints. We used three complementary methods to identify and map IESs.

1. MAC read alignment to MIC (*Figure 5—figure supplement 1A*). The previously generated Sanger sequencing reads from MAC genomic DNA libraries (*Eisen et al., 2006*) were mapped onto the entire MIC genome assembly using BWA (*Li and Durbin, 2010*) (RRID:SCR_010910). These mappings were used to detect 'split reads', adjacent parts of which mapped (in the same orientation) to two separate locations on the same MIC scaffold, putatively due to the presence of an IES in the region spanned by the read. To reduce false positive identifications, we required that every 'high confidence' IES be supported by at least three spanning reads, that the IES have low MAC read coverage (<1X, compared with overall average of 9.3X for MAC-destined regions; low coverage may occur due to MIC contamination of the original MAC libraries), and that the IES not begin or end with a sequencing gap.

2. MIC read alignment to MAC (*Figure 5—figure supplement 1B and C*). MIC genomic reads from this study were mapped to the MAC genome assembly using BWA (*Li and Durbin, 2010*) (RRID:SCR_010910). Duplicate reads due to PCR artifacts were removed using samtools-rmdup (http://samtools.sourceforge.net/samtools.shtml; RRID:SCR_002105). We identified reads whose alignment of stops before their ends (putatively because such reads contain IES sequence at their ends). At least six such 'broken' reads were required to validate a 'residual' IES site in the MAC genome. Also, we required that at least one broken read must face the residual site from each direction, with alignment breakpoints within 10 bp of each other. Most breakpoints from the 'left' or 'right' read direction were either adjacent or overlapped slightly, indicating a short direct repeat at the IES junction (*Figure 5—figure supplement 1C*).

3. MIC-MAC cross-assembly alignment. All MAC and MIC scaffolds were aligned to one another using nucmer (criteria: percent identity > 95, alignment length > 1000 bp; Mummer package RRID:SCR_001200). The presence of a putative IES was deduced by a larger gap (>100 bp difference) in adjacent alignment endpoints in the MIC genome than the corresponding endpoint gap in the MAC genome.

Applying stringent criteria, the first method identified 7757 putative IESs; nearly all were corroborated by one or both of the other two methods. Following manual curation, a final 'high confidence' set of 7551 IESs remained (*Supplementary file 3A*). An important criterion for inclusion in the high-confidence set is that the precise junction positions must be identified; internal assembly gaps are permitted, but gaps at IES/MDS junctions are not. In contrast to IES mapping method number one above, the other two mapping methods frequently identified putative IESs with terminal assembly gaps or, in many cases, consisting entirely of gap regions. These gaps are likely the result of two

factors: (1) repetitive sequences within the IESs and (2) minor contamination of the MIC sequencing libraries with MAC DNA (the resulting mixture of inconsistent short reads at IES/MDS junctions causing the assembly algorithm to introduce 'breaks'). Nevertheless, manual curation indicates that IESs likely exist at the great majority of these sites. Each of the latter two methods identified about 12,000 IESs in total, twice the number previously estimated based on extrapolation from a highly limited subset (*Yao et al., 1984*) or low-coverage MIC genome sequencing (*Fass et al., 2011*).

To confirm the validity of exceptionally short (<250 bp) or long (>25 kb) IESs, the read alignments were visually inspected using a JBrowse instance. IESs < 250 bp in length and three slightly larger coding region IESs were also confirmed by PCR amplification from a SB210 MIC genomic DNA sample with flanking primers using Q5 DNA polymerase (New England Biolabs, Ipswich, MA) or Platinum Taq HiFi DNA polymerase (Life Technologies-Thermo Fisher Scientific, Waltham, MA) according to manufacturer's recommendations. Expected sizes of MIC and MAC (from minor contamination of the template) amplification products were confirmed by gel electrophoresis. In one case where an exonic IES was within a sequencing gap, the MIC PCR product was cloned into the pCR4-TOPO vector (Life Technologies, Inc.) and sequenced on both strands.

## Repeat analysis

A library of repeats for the *T. thermophila* MIC genome was built by combining known transposable elements (*Jurka et al., 2005*), all previously described TE families in *Tetrahymena* (*Fillingham et al., 2004*; *Tsao et al., 1992*; *Wells et al., 1994*), and repeats identified de novo (Supplementary Dataset 1). To build the latter, we used RepeatScout (*Price et al., 2005*) (RRID:SCR_014653) with default parameters (>3 copies) to generate consensus repeat sequences. Those with greater than 90% sequence identity and a minimum overlap of 100 bp were assembled using Sequencher (v 4.07; RRID:SCR_001528). Repeats were classified into TE families using multiple lines of evidence, including detection of conserved TE protein domains (*Marchler-Bauer et al., 2011*), homology to known elements, presence of Terminal Inverted Repeats (TIRs), and detection of Target Site Duplications (TSDs). Homology-based evidence was obtained using RepeatMasker Protein Mask (*Smit et al., 2015*) (RRID:SCR_012954), as well as the homology module of the TE classifying tool RepClass (*Feschotte et al., 2009*) (RRID:SCR_014654). Repclass was also used to identify some of the signatures of transposable elements (TIRs, TSDs). We then eliminated non-TE repeats (simple repeats or gene families). First, consensus sequences were labeled as simple repeats or low complexity if 80% of their length could be annotated as such by RepeatMasker (masking of the library with the option –noint; RRID:SCR_012954). Next, consensus sequences were interrogated against protozoan refseq mRNAs (release 58) with tblastx (*Altschul et al., 1990*) (RRID:SCR_011823), and considered as non-TEs when: 1) evalue of the hit was lower than 1E-10; 2) the consensus sequence was not annotated as a TE; and 3) the hit was not annotated as a transposase. To facilitate this step, we used custom perl scripts available at https://github.com/4ureliek/ReannTE.

A selected subset of 32 of the 1674 resulting repeats was manually verified and consensus sequences were curated. Two non-exclusive criteria were used to select sequences for manual annotation: highest genome coverage or lowest divergence between copies (e.g. potentially recently active elements), based on a preliminary repeat annotation of the genome with RepeatMasker. Presumably, because of the large size and complex structure of *Maverick*/Tlr elements (*Pritham et al., 2007*), we could not identify any complete elements in the MIC assembly (the only copy that we could annotate within a unique scaffold was lacking some internal domains). Lastly, we further classified the remaining unclassified repeats (1480/1674), using a k-mer-based method and were able to assign classification for an additional 161 repeats. The use of this k-mer based tool (*Flygare et al., 2016*) (RRID:SCR_014655; available at https://github.com/Yandell-Lab/taxonomer_0.5) allowed us to test whether each copy of all unclassified repeats (query) could be classified as any copies of any classified repeats (database). The query's classification was updated to reflect the classification of the majority of its copies with significant scores. Repeats that showed inconsistent classification (such as DNA or LINE1 depending on copies) remained unclassified. Most repeats (1319) remain unclassified and may correspond to either TEs or to non-TE repeats: therefore, we refer to this custom library as putative TEs.

The *Tetrahymena* MIC and MAC genome assemblies were masked with RepeatMasker (*Smit et al., 2015*) using the refined repeat library (options –s and –nolow). Data presented in text and figures were obtained by parsing the RepeatMasker output file with custom perl scripts

(parseRM.pl and parseRM_GetLandscape.pl, https://github.com/4ureliek/Parsing-RepeatMasker-Outputs; *Figure 6—source datas 2* and *3*).

## Annotation of protein-coding genes

Structural annotation of MAC protein-coding genes was carried out independently at the Broad Institute and JCVI using an overlapping set of procedures. Differences in the results were resolved at JCVI. The strand-specific RNAseq data described above and non-strand-specific RNAseq data generated previously (*Xiong et al., 2012*) were assembled using Inchworm (*Grabherr et al., 2011*) (Trinity package, RRID:SCR_013048). These assemblies were aligned to the MAC genome using PASA (*Haas et al., 2003*) (RRID:SCR_014656). The gene structures of about 2700 inchworm assemblies with long open-reading frames (ORFs) and full-length gene structure support (i.e. a complete start-to-stop ORF preceded by an in-frame stop) were extracted and randomly split in half; one set was used for ab initio gene finder training while the other set was used to evaluate gene finder performance. Based on our previous experience with annotation of oligohymenophoran ciliate genomes (*Coyne et al., 2008*, *2011*; *Eisen et al., 2006*), we focused on the gene finders Genezilla (*Majoros et al., 2004*, *2005*) (RRID:SCR_014657) and Augustus (*Stanke and Morgenstern, 2005*) (RRID:SCR_008417). Parameters were iteratively modified by automated implementation of algorithmic scripts to optimize sensitivity and specificity of gene finder performance. We also searched the full genome assembly for protein homology and identifiable protein domains using the Analysis and Annotation Tool (AAT) package (*Huang et al., 1997*) (RRID:SCR_014658) against in-house, curated databases of non-redundant protein sequences (allgroup) and PFAM domains (TIGRFAM; RRID: SCR_005493). EVidence Modeler (EVM) (*Haas et al., 2008*) (RRID:SCR_014659) was used to combine RNAseq evidence, ab initio gene predictions and similarity evidence into initial gene model predictions. Gene models were updated using PASA and the RNAseq assemblies described above as well as approximately 60,000 previously generated, Sanger ESTs (*Coyne et al., 2008*). Omissions and structural differences between the three models (the new and previous [*Coyne et al., 2008*] JCVI annotations and the Broad Institute annotation) were assessed using GSAC (Gene Structure Annotation Comparison, unpublished), a JCVI in-house tool that evaluates coordinate differences between two gff3 (generic feature format version 3) files. Models present in either of the latter two annotations, but absent from the the new JCVI annotation, were evaluated and included in the final set if they met either of the following criteria: significant homology to a predicted gene in NCBI's non-redundant protein database (nr) or microarray evidence of transcription (*Miao et al., 2009*). Manual annotation steps were performed to correct cases of gene model overlap on the same or opposite strand, to edit or add genes curated on TGD (www.ciliate.org), and to correct translation of genes encoding selenocysteine-containing proteins (*Coyne et al., 2008*). Ab initio models that were fewer than 50 codons in length and lacked support from transcriptomic, protein homology or domain evidence were deleted. Functional gene product names were assigned by implementation of a JCVI pipeline designed to weigh multiple sources of sequence evidence, select the best supported names, and homogenize nomenclature within paralogous families.

To evaluate the protein-coding potential of MIC-limited genomic regions, we applied similar methods to the set of 5625 fully sequenced IESs and 21 NMCs > 500 bp in length. Gene-finding algorithms Augustus and Genezilla, trained on MAC genome data as described above, were applied to generate ab initio gene predictions. Unfortunately, the available RNAseq reads were often unable to validate and improve these predictions. In a separate study, Gao et al. (unpublished) used RNAseq evidence from a specific time point during conjugation to model a number of MIC-limited genes located within both IES and NMC regions. Full characterization of the IES transcripts will be presented in a later paper. For this study, genes within NMCs were hand-curated for maximum reliability. Many of these NMC models also had RNAseq support within our available evidence (see *Supplementary file 2G*).

## Small RNA analyses

Small RNA purification, co-immunoprecipitation, and analyses by denaturing gel electrophoresis were performed as previously described (*Noto et al., 2010*). Construction of small RNA cDNA libraries, high-throughput sequencing, and data processing were performed as previously described (*Mochizuki and Kurth, 2013*; *Noto et al., 2014*; *Schoeberl et al., 2012*). The data for Twi1p- and

Twi11p-bound small RNAs have been deposited at the NCBI Gene Expression Omnibus (www.ncbi.nlm.nih.gov/geo/) as GSE79849 and GSM1672144, respectively.

## IES excision variability

To assess the degree of variability in IES excision endpoints, we sequenced genomic DNA from a pool of MACs derived from independent differentiation events. We mated the sequenced reference strain SB210 to SB1969 (RRID:TSC_SD00701). Both are whole-genome homozygous cell lines of *T. thermophila* inbred strain B (*Allen and Gibson, 1973*). Many fissions after they last conjugated, both strains were subcloned from a single cell, ensuring that essentially every MAC locus has become homozygous through phenotypic assortment, eliminating parental MAC IES excision endpoint variation; our sequence analysis verified this prediction. Mating exconjugants were selected for resistance to cycloheximide (the allele for which is found in the MIC genome of SB1969 and only expressed in true progeny) in 96-well plates at a dilution that limited the number of independent resistant progeny pairs in each well to an average of two. After 3 days growth in cycloheximide, resistant cells from 330 wells were pooled. This pool contained the vegetative descendants of about 660 mating pairs, or 1320 individual progeny. In each progeny cell, two MACs develop and undergo programmed genome rearrangement; thus, the pool represents about 2640 independently derived MACs, but given that IES excision occurs when the MAC ploidy is about 8°C, the diversity of excision endpoints is potentially even greater. To expand the cell population for MAC DNA purification, the pooled cells were grown for approximately eight cell divisions in a large volume of medium to minimize competition. MACs from the progeny pool, as well as both parental strains, were purified by standard procedures (*Gorovsky et al., 1975*) and DNA prepared from the nuclei. Illumina (San Diego, CA) sequencing libraries, with a fragment size of about 600 bp, were prepared from each DNA sample and sequenced (2 × 100 base paired-end reads) using an Illumina HiSeq2000 instrument following manufacturer's recommendations. MAC genome coverage of each sample was: SB210 = 27X, SB1969 = 66X, Progeny Pool = 156X. These MAC reads were mapped to the SB210 MIC genome scaffolds to identify 'split reads', as described above.

## Data deposition and reagent availability

The sequenced strain is available from the *Tetrahymena* Stock Center (https://tetrahymena.vet.cornell.edu/; Stock ID SD01539). All Illumina RNA and DNA sequence data were submitted to the NCBI Short Read Archive (SRA) (http://www.ncbi.nlm.nih.gov/sra) and can be retrieved using the following accession numbers: DNA BioProject PRJNA51571, RNA BioProject PRJNA177770. Micronuclear genome assembly sequences has been deposited at DDBJ/ENA/GenBank under the accession AFSS00000000. The version described in this paper is version AFSS02000000. The macronuclear whole genome shotgun project has been deposited at DDBJ/EMBL/GenBank under the accession AAGF00000000. The macronuclear genome annotation update described in this paper is version AAGF03000000. The information presented in this paper is also available at http://datacommons.cyverse.org/browse/iplant/home/rcoyne/public/tetrahymena/MIC and in a browser (JBrowse format at http://www.jcvi.org/jbrowse/?data=tta2mic). The browser 'Golden Path' shows each MIC chromosome super-assembly and its relationship to Cbs's, IESs, MAC genes, MAC scaffolds, MIC specific gaps, and NMC genes.

## Acknowledgements

We thank the Broad Genomics Platform for their contribution and support. We thank James Bochicchio for project coordination, Margaret Priest for annotation and Lucia Alvarado-Balderrama for data release. This work was supported by the National Human Genome Research Institute grant U54 HG003067 (to Eric S Lander and Stacey Gabriel), by award MCB-1158346 from the National Science Foundation to RSC, and by award 31525021 from the Natural Science Foundation of China to WM, and by the National Institutes of Health grant GM077582 to CF. We thank the following Southern Illinois University students for technical assistance (listed in chronological order of their contributions): Courtney Taylor, Kylie Corry, Logan Roberts, Benjamin Clevenger, Nicole Szczepanik, Brian Pinkins, and Aparajita Rajamahanty. We thank the following University of California at Santa Barbara students for technical assistance (listed in alphabetical order): Sergio Alvarez, Brian Argueta, Justin Kanerva, Gavriel Matt, Veronica Munoz, Atul Saini, Richard Sanchez, and Leinah Tran, as well as technician

Judith Orias. We thank the following Cornell University students: Francis Chen, Shanique Alabi, Kacey Solotoff, and Gary Tan, as well as technician Mozzamal Hossain. We gratefully acknowledge Chris Town for administrative management and helpful discussions, Heather B. McDonald for scientific illustration, and the Next Generation Sequencing unit of Campus Support Facility, Vienna Bio-Center, for sequencing small RNAs.

## Additional information

### Funding

| Funder | Grant reference number | Author |
| --- | --- | --- |
| National Institutes of Health | GM077582 | Cédric Feschotte |
| Natural Science Foundation of Hubei Province | 31525021 | Wei Miao |
| National Science Foundation | MCB-1158346 | Robert S Coyne |

The funders had no role in study design, data collection and interpretation, or the decision to submit the work for publication.

### Author contributions

EPH, EO, RSC, Conceived and designed study, Experimental work and data analysis, Drafted and revised article; AK, Bioinformatic data analysis, Drafted and revised article, Acquisition of data ; PEH, Molecular genomic analysis, Drafted sections of article, Acquisition of data; SLB, NZ, HT, MH, VK, JT, CMC, Bioinformatic analysis, Acquisition of data; JHB, Phylogenetic analysis; EVC, EJP, CF, Bioinformatic analysis; CR, Genome sequencing, Assembly, and Annotation, Conception and design, Acquisition of data, Analysis and interpretation of data, Drafting or revising the article; QZ, LF, JZL, TS, SKY, RH, RD, SG, JRW, Genome sequencing, Assembly, and Annotation, Acquisition of data, Analysis and interpretation of data; BWB, CN, Genome sequencing, Assembly, and Annotation, Analysis and interpretation of data; TN, Bioinformatic analysis, Molecular biology experiments; KM, Bioinformatic analysis, Molecular biology experiments, Drafting or revising the article; RP, SDT, Generation of micronuclear DNA, RNA, Contributed unpublished essential data or reagents; PHD, Genetic mapping, Acquisition of data, Analysis and interpretation of data; DMC-H, Genetic deletion mapping, Acquisition of data, Analysis and interpretation of data; JX, WM, Generation of RNAseq data, Acquisition of data, Analysis and interpretation of data, Contributed unpublished essential data or reagents

### Author ORCIDs

Aurélie Kapusta, http://orcid.org/0000-0002-4131-903X
Vivek Krishnakumar, http://orcid.org/0000-0002-5227-0200
Kazufumi Mochizuki, http://orcid.org/0000-0001-7987-9852
Robert S Coyne, http://orcid.org/0000-0002-7693-3996

## Additional files

### Supplementary files

• Supplementary file 1. Genome assembly and centromere structure. (A) Genome assembly statistics. (B) T. thermophila superscaffolds assembled by HAPPY physical mapping. (C) Chromosome super-assemblies. (D) Suspected Chimeric MIC Supercontigs. (E) Most centric MAC chromosomes are among the ten longest MAC chromosomes.

• Supplementary file 2. Chromosome breakage and NMCs. (A) The 225 Cbs identified in the T. thermophila MIC genome. (B) The 181 T. thermophila MAC chromosomes: Lengths and flanking Cbs's. (C) Conservation of chromosome breakage sites 3L-15 to 26 relative to flanking genes in all four examined Tetrahymena species. (D) Identity and lengths of Tetrahymena species scaffolds and supercontigs defined by Cbs's homologous to T. thermophila 3L-26 to 15. (E) T. thermophila Cbs

clade alignments. (F) Properties of predicted non-maintained chromosomes. (G) Predicted genes on non-maintained chromosomes.

• Supplementary file 3. Internally eliminated sequences. (A) 7551 High Confidence IESs. (B) Summary of the putative TE content in IESs. (C) Interspersed repeat (putative TEs) content. (D) Characteristics of IESs within protein-coding regions.

• Supplementary file 4. Cbs segment sequences and alignments.

## Major datasets

The following datasets were generated:

| Author(s) | Year | Dataset title | Dataset URL | Database, license, and accessibility information |
|---|---|---|---|---|
| Russ C, Coyne RS, Orias E, Taverna SD, Papazyan R, Young SK, Zeng Q, Gargeya S, Fitzgerald M, Haas B, Abouelleil A, Alvarado L, Arachchi HM, Berlin A, Brown A, Chapman SB, Chen Z, Dunbar C, Freedman E, Gearin G, Goldberg J, Griggs A, Gujja S, Heiman D, Howarth C, Lui A, MacDonald PJP, Montmayeur A, Murphy C, Neiman D, Pearson M, Priest M, Roberts A, Saif S, Shea T, Sisk P, Stolte C, Sykes S, Wortman J, Nusbaum C, Birren B | 2014 | Micronuclear Genome Assembly | http://www.ncbi.nlm.nih.gov/nuccore/AFSS00000000 | Publicly available at NCBI Nucleotide (accession no: AFSS00000000) |
| Russ C, Coyne RS, Orias E, Taverna SD, Papazyan R, Young SK, Zeng Q, Gargeya S, Fitzgerald M, Haas B, Abouelleil A, Alvarado L, Arachchi HM, Berlin A, Brown A, Chapman SB, Chen Z, Dunbar C, Freedman E, Gearin G, Goldberg J, Griggs A, Gujja S, Heiman D, Howarth C, Lui A, MacDonald PJP, Montmayeur A, Murphy C, Neiman D, Pearson M, Priest M, Roberts A, Saif S, Shea T, Sisk P, Stolte C, Sykes S, Wortman J, Nusbaum C, Birren B | 2014 | Tetrahymena thermophila SB210 micronuclear genome sequencing | http://www.ncbi.nlm.nih.gov/bioproject/PRJNA51571/ | Publicly available at NCBI BioProject (accession no: PRJNA51571) |

| Bidwell S, Michalis HM, Zafar N, Joardar V, Miao W, Russ C, Eisen J, Wu M, Wu D, Nierman W, Orias E, Delcher A, Salzberg S, Coyne R | 2014 | Macronuclear Genome Assembly | http://www.ncbi.nlm.nih.gov/nuccore/AAGF00000000 | Publicly available at NCBI Nucleotide (accession no: AAGF03000000) |
|---|---|---|---|---|
| Russ C, Coyne RS, Orias E, Taverna SD, Papazyan R, Young SK, Zeng Q, Gargeya S, Fitzgerald M, Haas B, Abouelleil A, Alvarado L, Arachchi HM, Berlin A, Brown A, Chapman SB, Chen Z, Dunbar C, Freedman E, Gearin G, Goldberg J, Griggs A, Gujja S, Heiman D, Howarth C, Lui A, MacDonald PJP, Montmayeur A, Murphy C, Neiman D, Pearson M, Priest M, Roberts A, Saif S, Shea T, Sisk P, Stolte C, Sykes S, Wortman J, Nusbaum C, Birren B | 2014 | Tetrahymena thermophila Transcriptome Sequencing | https://www.ncbi.nlm.nih.gov/bioproject/PRJNA177770/ | Publicly available at NCBI BioProject (accession no: PRJNA177770) |

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

# Appendix 1. Observations and hypotheses relevant to Cbs duplication and evolution in tetrahymenine ciliates

## Cbs duplication clades: three case histories

Based on a statistically significant degree of sequence similarity, at least 15 Cbs segment clades have been identified in the *T. thermophila* MIC genome, as described in the main text, *Table 3*, and *Supplementary file 2E*, above. Below we consider in more detail three such clades, which collectively illustrate important features of the duplication process that generates Cbs clades. The case histories are parsimonious in that they minimize the number of distinct genetic events required to explain the diversity observed; they are only meant to illustrate the type of events most likely involved.

### The Cbs 1 L-1 clade: a tandem Cbs repeat clade with short repeat periodicity

This clade occupies a continuous 726 bp MIC DNA segment and consists of six tandemly repeated copies of ~120 bp sequence that contains a Cbs (sequence and alignment shown in *Supplementary file 4*, section A1; alignment statistics in *Appendix 1—table 1*). A phylogenetic tree of the six repeat units, obtained as described under Materials and methods, is shown in *Appendix 1—figure 1*. A plausible, simple sequence of duplications and Cbs mutations, consistent with both the phylogenetic tree of the repeats and their order along the MIC chromosome, is illustrated in *Appendix 1—figure 2*. Some comments on this case history follow.

**Appendix 1—table 1.** Match statistics of all-by-all Blastn alignments of Cbs 1L-1 clade members.

| Cbs | Len* | 1L-1 | 1L-2 | 1L-3 | 1L-4 | 1L-AAC | 1L-5 |
|---|---|---|---|---|---|---|---|
| 1L-1 | 94 | −48 | −29 | −22 | −23 | −29 | −25 |
| 1L-2 | 120 | 83:74%,0% | −62 | −45 | −43 | −43 | −40 |
| 1L-3 | 122 | 96:80%,5% | 122:89%,3% | −63 | −49 | −45 | −41 |
| 1L-4 | 118 | 82:69%,0% | 118:89%,0% | 122:92%,3% | −61 | −40 | −39 |
| 1L-AAC | 121 | 93:80%,4% | 119:89%,2% | 123:89%,5% | 119:87%,2% | −62 | −49 |
| 1L-5 | 118 | 90:84%,1% | 116:88%,0% | 120:88%,3% | 116:87%,0% | 121:93%,2% | −61 |

* Len = length; 1L-AAC - non-functional Cbs variant with AAC insertion.

Main diagonal: Self-match E-values (highlighted yellow). Cells above main diagonal: Expected value of non-self matches. Cells below main diagonal: match length: % sequence identity, % gaps.

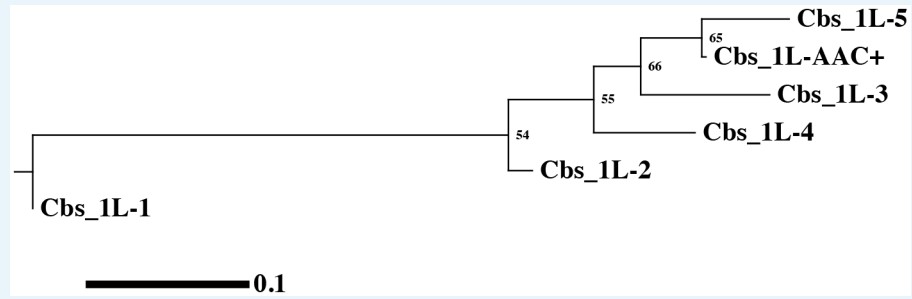

**Appendix 1—figure 1.** Phylogenetic tree of the 1L-1 clade. Phylogenetic tree of the 1L-1 clade. The branches show significant statistical support, as indicated by bootstrap percentages.

**Appendix 1—figure 2.** A possible history of the Cbs 1L-1 clade. Cbs 1L-+ represents the nonfunctional variant with an internal trinucleotide insertion. Line 1: the putative ancestral Cbs and adjacent sequence. Line 10: final (current) state of the 1L-1 clade. Divergent pair of arrows: repeat unit duplication. Crossed arrows: circular permutation of two repeat units. Vertical single arrow: Cbs mutation. Generation of duplications and the circular permutation by unequal crossing-over is diagrammed in *Appendix 1—figure 3*.

a. The relative ease with which duplications, deletions, circular permutations and gene conversions can occur in tandem repeats suggests that the duplication history in any tandem repeat clade is likely to be complex. Thus, only a minimum, representative number of events can be inferred from the current state of the repeats.

b. The generation of the first two tandem repeats, containing a substantial length of repeated sequence, is likely the rate-limiting step in the expansion because the first duplication likely depends on microhomology-dependent non-homologous end joining. Additional duplications (and deletions) of tandem repeat units are likely to follow with higher frequency due to unequal crossing-over, i.e., non-allelic homologous recombination, greatly facilitated by the presence of neighboring repeat copies that can align out of register with one another (see *Appendix 1—figure 3*). These events must occur in the MIC, either during meiosis or as a result of DNA damage repair during vegetative multiplication.

c. Because unequal crossover generates a pair of reciprocal products, representing duplication and deletion, the number of repeats in a clade is elastic during evolutionary time. Thus, the current number of repeats may be but a snapshot of a dynamic process and the history shown in *Appendix 1—figure 3* may represent just one among many possibilities.

d. Duplications are generated as immediately adjacent pairs in the simplest unequal cross-over model (*Appendix 1—figure 3A*). The order of forks in the phylogenetic tree (*Appendix 1—figure 1*) leaves the repeats in scrambled order, 1,2,4,3,+,5 instead of 1,2,3,4,+,5, differing by a circular permutation of repeats 3 and 4. *Appendix 1—figure 3B*

shows how a series of duplications/deletions caused by unequal crossovers can generate the circular permutation. On the other hand, the branching of the tree may be incorrect, as the bootstrap values are not very high. Furthermore, the recombination junctions in successive unequal crossover events need not occur at equivalent nucleotide locations; this may decrease statistical support for the phylogenetic tree and blur its relationship to the physical order of the repeats.

e. Two distinct mutation events – the minimum number required to generate the three functional Cbs variants observed in this clade – are invoked (*Appendix 1—figure 2*). The clade ancestor is assumed to have had the Cbs 11C variant as that is the possibility most simply consistent with the repeat history embodied in the phylogenetic tree. These mutations become convenient markers for the repeats.

f. Interestingly, the Cbs in the fifth repeat copy has mutated to a variant having an AAC insertion between Cbs positions 7 and 8. The Cbs is almost certainly non-functional, as the two positions are invariant among nearly 200 functional Cbs's that define the ends of the maintained MAC chromosomes. The 2R-1 clade is a tandem repeat containing three repeat units, averaging 311 bp per repeat unit. Every Cbs shares the 14A substitution. Similar to the 1L-1 clade, the middle repeat contains a presumably nonfunctional Cbs with a 9T substitution. Thus, Cbs's are subject to 'birth' (duplication) and 'death' (mutation to a non-functional variant) evolutionary events.

g. Because the predicted MAC chromosomes defined by these Cbs's are non-maintained (NMCs; *Supplementary file 2F*, main text), none of these duplication events would have affected the number of maintained chromosomes in the MAC genome. Furthermore, the repeat length is very short and there are no predicted genes within these repeats. This raises the question of why this clade contains five Cbs's that have retained functionality when just one would suffice. The simplest answer would be that the duplication events occurred very recently, a conclusion supported by the high degree of Cbs-flanking sequence conservation observed (see alignment in *Supplementary file 4*, section A1).

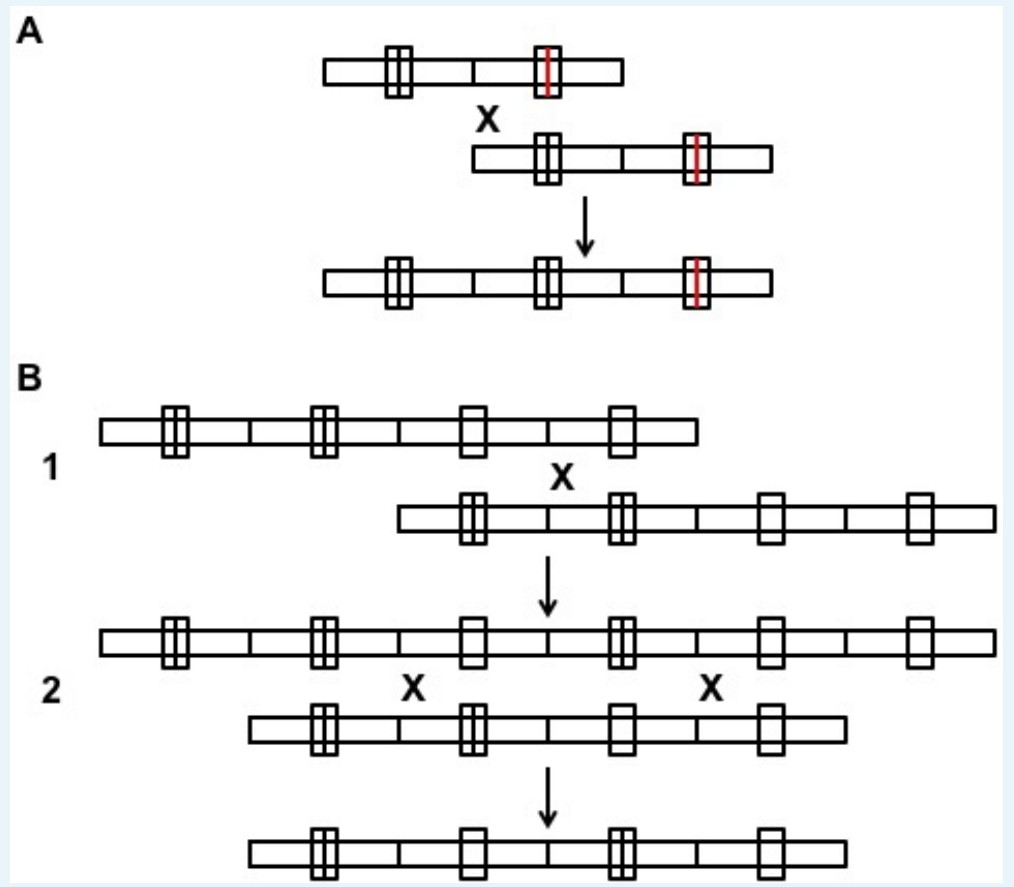

**Appendix 1—figure 3.** Examples of unequal crossing over. (**A**) Repeat unit duplication. (**B**) Circular permutation of two adjacent repeats. X: unequal crossing over by non-allelic homologous recombination; only the recombinant product of interest is shown. The circular permutation shown involves a series of two independent unequal cross-overs, one of which is a double cross-over; the latter step could alternatively be replaced with two serial single cross-overs (not shown). Another alternative, starting with the original 4-repeat sequence, is a unimolecular unequal cross-over that excises a circle containing the two middle repeats (not shown). Immediate re-insertion of the circle by unequal crossing over at the circle location diametrically opposed to that of the excision site, would accomplish the identical circular permutation more economically.

## The Cbs 1R-35 and 1R-37 clades: members of a putative superclade

These two clades are consecutive, and each contains two members (see *Supplementary file 4*, section A2 for sequences and alignments). A possible evolutionary history of the four Cbs's that explains the known features of these four Cbs's segments is shown in *Appendix 1—figure 4*. The putative superclade founder Cbs is proposed to be the 1A,11C variant because that requires the minimum number of Cbs mutational events within the entire proposed superclade: a 1A,11C to 1A mutation. The final duplications created two predicted NMCs and thus would not have changed the number of maintained MAC chromosomes.

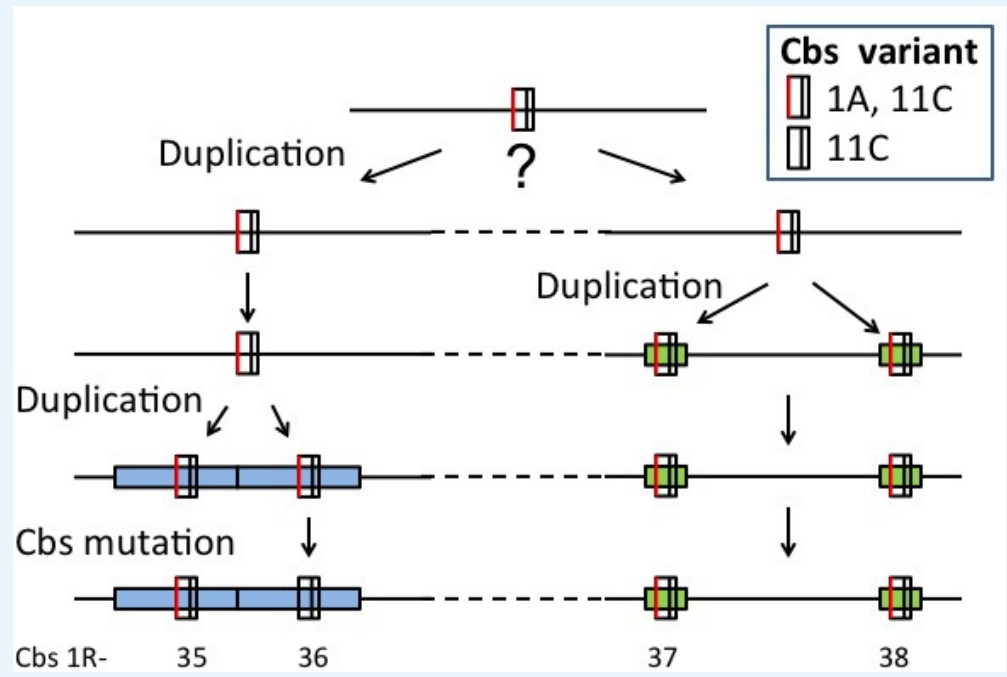

**Appendix 1—figure 4.** A putative superclade encompassing the Cbs 1R-35 and 1R-37 clades. The Cbs in these two clades are consecutive (bottom line). The top line represents the putative ancestral Cbs. The lengths of the alignments are 512 bp (blue shading) and 150 bp (green shading) for Cbs 1R-35/36 and Cbs 1R-37/38, respectively.

These considerations suggest the existence of additional superclades, whose relationship has been erased by the accumulation of random mutations in Cbs-adjacent regions during evolutionary time.

### The Cbs 1L-16 and 1L-17 clades: Evidence for a simultaneous duplication/translocation of the rDNA chromosome sequence and flanking Cbs's

The 3' end of the ~11 kb nascent rDNA MAC chromosome in *T. thermophila* is defined by a single Cbs, 1L-16, whereas three closely spaced, tandemly repeated Cbs's – 1L-17, 18 and 19 – define its 5' end, with Cbs 1L-17 being the rDNA-proximal Cbs (*Appendix 1—figure 5*). These Cbs's were the first to be identified (*Yao et al., 1985*). The MIC sequence corresponding to the 5'end of the nascent rDNA chromosome contains an inverted pair of identical 42 bp repeats, the M-repeats, separated by a 28 bp single-copy non-palindromic DNA sequence (*Yao et al., 1985*) (*Appendix 1—figure 5*). The nascent rDNA MAC chromosome is rearranged by a programmed, unimolecular, homologous recombination and repair event that generates a ~21 kb palindrome – the mature form of the rDNA chromosome (*Butler et al., 1995*) (see *Appendix 1—figure 5*). The inverted M-repeats are required for palindrome formation (*Yasuda and Yao, 1991*). The palindromic configuration of the mature MAC rDNA and the M-repeat sequence are conserved in every examined species of genus *Tetrahymena*, whereas – with the exception of two closely related species (*Coyne and Yao, 1996*; *Engberg, 1983*) – the sequence of the non-palindromic segment separating the M-repeats is not conserved. Interestingly, at least in *T. thermophila*, the *specific sequence* of the M-repeats is not required for rDNA palindrome formation; they retain their function when replaced with inverted repeats of unrelated sequence (*Yasuda and Yao, 1991*).

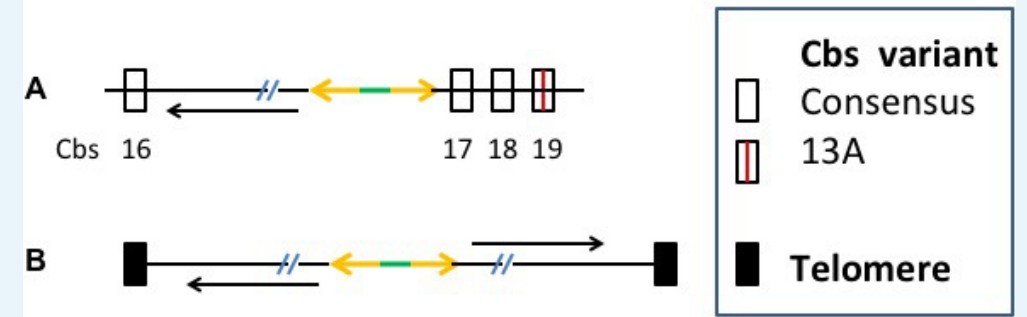

**Appendix 1—figure 5.** Cbs in the 1L-16 and 17 clades flank the MIC rDNA chromosome-destined DNA. (**A**) The ~11 Kb MIC form of the rDNA MAC chromosome-destined DNA. 5'and 3' ends, as defined with respect to the rRNA coding region, are on the right and left, respectively. Rectangles: flanking Cbs; orange arrows: inverted 42 bp M-repeats; green segment: 28 bp M-repeat non-palindromic spacer. Diagonal slashes: rDNA segments, including the rRNA gene, not shown. (**B**) The mature, palindromic MAC rDNA chromosome. Black rectangles: Telomeres. Long black arrows in both panels indicate the 5' to 3' direction of the coding strand of the rRNA gene.

Our super-assembly of chromosome 4 revealed strong evidence for a duplication of the MIC region containing Cbs 1L-17 to 19 and its inter-chromosomal translocation to MIC chromosome 4R. The matching segment in chromosome 4R includes one functional Cbs, 4R-25, as well as 381 bp which matches a segment at the 5' end of the MAC rDNA, including the 112 bp region containing the M-repeat copies and their non-palindromic spacer (see alignment and sequences in *Supplementary file 4*, section A3). The duplicated/translocated sequence most likely had only 2 Cbs's at this end, as the expected value of a chance BlastN alignment further decreases by at least 4 orders of magnitude if the 45 bp 1L-17 repeat unit is deleted (data not shown). Given the nearly complete sequence identity of the Cbs 1L-17 and 18 repeat units, it seems likely that they were generated by a more recent duplication than the 1L-4R duplication/translocation event. Cbs 4R-25 is probably homologous to Cbs 1L-19 because both share a 13A substitution. The 13A substitution is not seen in Cbs 1L-18 or the degenerate (non-functional) Cbs that occupies the corresponding location on 4R (compare the MIC sequences of the 5' rDNA region and the region including Cbs 4R-24 and 25). Remarkably, the sequence adjacent to the Cbs at the other (3') end of the rDNA sequence, Cbs 1L-16, matches the sequence adjacent to Cbs 4R-24, the nearest neighbor of Cbs 4R-25, separated from it by only 656 bp (see alignment of Cbs 4R-24 and 3' rDNA end and their sequences in *Supplementary file 4*, section A3). The matching sequence includes 129 bp of sequence similar to the non-transcribed segment at the 3' end of the rRNA gene.

It is very likely that the entire MIC rDNA chromosome sequence and flanking Cbs's were translocated in a single event. As can be seen in the sequence (*Supplementary file 4*, section A3), the segments that match the rDNA 5' (dark gray) and 3' (light gray) ends are only 27 bp apart. If the duplication/translocation went from 1L to 4R, the probability of two independent duplications/translocations ending up 27 bp apart in a >150 Mb genome are remote. Conversely, if independent duplication/translocations went from 4R to 1L, and given that the M-repeats sequence is dispensable for the cell but their inverted deployment is required for rDNA palindrome formation, it is not easy to explain why both segments were in 4R, unless they already flanked a complete, functional copy of the rDNA. Consistent with the single translocation hypothesis, the homologous Cbs's have the same orientation with respect to one another and Cbs duplication-and-translocations to a different chromosome tend to be rarer than to the same chromosome (main text, *Supplementary file 2E*).

These considerations lead us to conclude that the entire ~11 kb rDNA segment, including the flanking Cbs's, was duplicated and translocated from MIC chromosome 1R to 4R (or vice versa). Subsequently, an internal segment, representing most of the rDNA, was deleted from the 4R copy. The excision site can be pinpointed to the 27 bp gap between the 5' and 3' matching segments in the sequence interval between Cbs's 4R-24 and 25 (see sequence in **Supplementary file 4**, section A3). Since MAC rDNA palindrome copy number is regulated at nearly 10,000 copies per cell (**Pearlman et al., 1979**), neither the putative rDNA gene duplication nor its subsequent deletion need have affected final rDNA copy number in the MAC. Furthermore, since a naturally occurring mutation can cause differential replication/maintenance of one allelic form and complete replacement of the other form in heterozygotes (**Pan et al., 1982**), a redundant copy could readily become entirely superfluous.

The simplest sequence of duplication/ translocation and Cbs mutation events that accounts for the current organization at both MIC chromosomes is shown in **Appendix 1—figure 6**. MIC DNA sequencing of additional tetrahymenine species should illuminate the interesting questions of eukaryotic chromosome and genome evolution raised by our observations.

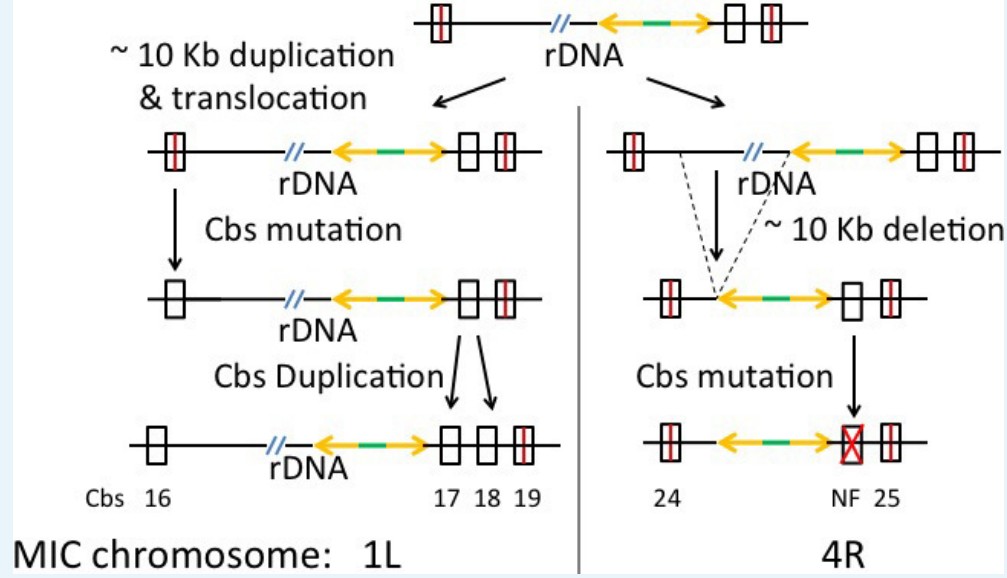

**Appendix 1—figure 6.** Cbs 1L-16 and 17 Clade: Simplest duplication, translocation and Cbs mutation history. Crossed rectangle labeled NF: Non-functional mutant Cbs; other symbols as in **Appendix 1—figure 5**. The top line is the putative ancestral rDNA region, in either chromosome 1L or 4R. The bottom line represents the current state of the duplicated/ translocated sequences.

Interestingly, in the tetrahymenine species *Colpidium campylum* and *Glaucoma chattoni*, the Cbs at the 3'end of the rDNA has opposite orientation to that of 1L-16, the corresponding Cbs in *T. thermophila* (**Coyne and Yao, 1996**). It follows that this difference must have arisen after these two species lineages diverged from the *T. thermophila* lineage. The same must be true for the 1R-4R rDNA duplication described above, as Cbs 4R-24 shows the same orientation with respect to the remnants of the flanking rDNA homologous DNA sequence. Not enough MIC sequence is available from the two other species to determine if the resulting inversion was an 'in situ' inversion or whether some other duplicated Cbs was translocated to the sequence adjoining the rDNA 3'end. Partial MIC sequences, adjacent to the 5' end of the rRNA gene, are available for two *Tetrahymena* species (*T. pigmentosa* and *T. hegewischi*) (**Coyne and Yao, 1996**); no evidence of sequence similarity is observed in this region between these species and

*thermophila.* By analogy to the 3' end Cbs, it seems very likely that the Cbs tandem repeat duplication in *thermophila* occurred after its divergence from the *pigmentosa* and *hegewischi* lineages.

## On the evolution of Cbs-mediated chromosome breakage in tetrahymenine ciliates

Although ciliate chromosome breakage is universally conserved, the tetrahymenine Cbs-mediated mechanism has not been found in any other ciliate group studied. Furthermore, there has been little divergence in the Cbs sequence itself within the tetrahymenine lineage (*Coyne and Yao, 1996* and this study). It is therefore likely that the Cbs-mediated mechanism for chromosome breakage was introduced fairly recently into the tetrahymenine lineage, after its divergence from that of ciliates in closely related groups, such as *Paramecium*. Additional lines of evidence consistent with this view have been presented in main text. How could the Cbs-mediated mechanism have taken over an ancestral mechanism of chromosome breakage?

As discussed in main text, there are strong reasons to believe that preservation of both the extent and locations of chromosome breakage in tetrahymenine ciliates are under purifying selective pressure. The tetrahymenine branch diverged from ciliates that already had highly evolved MAC biology, including programmed chromosome breakage, and by inference, an ancestral chromosome breakage system already optimized by natural selection. This suggests that the spread of Cbs's may have been largely restricted to pre-existing breakage sites. In any case, rampant spread of breakage sites in the germline would certainly be deleterious to progeny survival. Transposons often display adaptations that avoid damaging host fitness; for example, the yeast Ty1 transposon inserts almost exclusively at "safe" sites upstream of promoters for RNA polymerase III-transcribed genes, by virtue of its targeting through association with a subunit of that enzyme (*Bridier-Nahmias and Lesage, 2012*). It is conceivable that the invading Cbs mobile element was targeted to preexisting sites of chromosome breakage by association of its transposase with one of the recognition factors involved in that ancestral process. Of course, in order to be heritable, this targeting would need to occur in the germline, where breakage does not normally occur, but perhaps the putative association drew the complex from its normal site of action, in the macronuclear anlage, into the germline nucleus, much as an HIV pre-integration complex can traverse the interphase nuclear membrane (*Bukrinsky, 2004*). Once in the micronucleus, mobile element integration at multiple pre-existing chromosome breakage sites might proceed (without resulting in germline chromosome breakage). In time, the Cbs endonuclease would be domesticated and all traces of the original mobile element, with the exception of the 15 bp Cbs, would degenerate by random mutation. Presuming a selective advantage for the highly precise and reliable Cbs-mediated breakage mechanism over the ancestral state, the new state would become fixed in the population.

Ultimately, validation or rejection of this speculative model will rest on the discovery of the endonuclease complex responsible for Cbs-mediated chromosome breakage. It is notable that a homolog of a telomerase subunit, Pot2, was recently found to associate with chromosome breakage sites in *T. thermophila* (*Cranert et al., 2014*). It may take part in the telomerase-dependent healing of the newly broken ends. Perhaps it may serve as the bait by which other components of the breakage complex may be identified. Alternatively, screening of homing endonuclease- and transposase-related genes in the *Tetrahymena* genome may trigger discovery of an association with programmed chromosome breakage.

