## [Decision Letter]

Thank you for submitting your work entitled "Structure of the germline genome of *Tetrahymena thermophila* and its relationship to the fragmented somatic genome" for consideration at *eLife*. Your revised article has been favorably evaluated by Diethard Tautz (Senior editor), a Reviewing editor (Kathleen Collins), and three reviewers.

All reviewers agree that the manuscript provides many interesting insights about what sequence comparison between Mic and Mac tells us about *Tetrahymena* biology, genome rearrangements, and evolution. The interesting observations include the presence of non-maintained chromosomes, genome un-scrambling, and the evolution of chromosome breakage sites (CBS), and the size, nature and elimination of the Mic centromeric repeated DNA.

The manuscript has some remaining issues that need to be addressed before it would be considered as accepted, as outlined below. In particular, all three reviewers and the Reviewing Editor believe that the work will have its highest impact if it is more readable to general audience of *eLife*. The text body could have provided a bit more guidance for readers as to the general take-home implications of their findings, as detailed below. In addition, the manuscript would benefit from having a summary figure illustrating the major findings: repeat rich centromeric regions, conserved CBS positions, and non-maintained chromosomes.

1) Much of the information from the subsections “MIC Genome Sequencing and Assembly” and “Building and Analysis of Chromosome-Length MIC Super-Assemblies” could be stated in a summary fashion in the text and the details that are presented could be put in the Methods or in supplemental material. This would help the paper read more smoothly.

2) The scrambled mac destined DNA described in the subsection “Scrambled Macronuclear Chromosome-Destined DNA” is of interest and the authors describe the mechanisms as different from what is in hypotrichs. It would be helpful to have a figure to follow what the authors have found. They state another manuscript will address this, but if they are going to report this at least to some degree here, a figure and a bit more detail about what is different would help a lot.

3) The first six paragraphs of the subsection “MIC Centromeres” could be shortened and condensed. The whole section is particularly long and speculative, and in places repetitive. The concept of meiotic drive in inserted in the middle of the description of the structure and comparison to other organisms. Can the section be edited to tighten up the prose and perhaps shortened?

4) The discussion of the programed elimination of Cen sequences in the sixth paragraph of the subsection “MIC Centromeres” needs to have a figure to allow the reader to follow the structures that are being referred to.

5) The seventh paragraph of the subsection “MIC Centromeres” discusses non-maintained chromosomes, which is discussed later. This does not belong under the heading "MIC Centromeres" where it currently is.

6) In the ninth paragraph of the subsection “MIC Centromeres”, the discussion of transcripts from mic sequences is really a summary of the literature and a hypothesis and should be in a dedicated discussion, not in the middle of a section on centromeres.

7) The discussion of Cbs's in the subsections “The Chromosome Breakage Sequence (Cbs) Family”, “The Set of 225 Cbs’s in the MIC Genome” and “Degeneracy of the Cbs motif among functional Cbs’s”, should be significantly condensed. The consensus sequence was already known, so to authors should spend less time of this, and just start the new information that the additional sequences offer.

8) Likewise the discussion of the conservation of CBS's across species in the subsections “Conservation of chromosome breakage sites across *Tetrahymena* species” and “Duplication of Cbs Regions on an Evolutionary Time Scale”, should be significantly shortened and combined with the identification of CSB sites discussed above.

9) The subsection “The Non-Maintained MAC Chromosomes” could be combined with the discussion of phenotypic assortments in the tenth paragraph of the subsection “MIC Centromeres” and this section should be shortened.

10) The summary and conclusion would not be needed if the authors state the results and then wrote a concise Discussion.

11) On Figure 2, it seems like it might be helpful if the authors denoted the region which they consider to contain the centromere of each chromosome.

12) In the fourth paragraph of the subsection “Duplication of Cbs Regions on an Evolutionary Time Scale”, it needs a period after "relatedness".

13) In the fourth paragraph of the subsection “Duplication of Cbs Regions on an Evolutionary Time Scale”. The idea about the CBS being introduced by a transposon invasion leading to the use of this sequence as the chromosome fragmentation site is compelling, but the way it is described is rather non-committal. It would not be "related" to a transposon invasion, it would result from the invasion and subsequent domestication of the breakage enzyme. This is one example related to my general comment that the manuscript could be written more succinctly and clearly.

14) The description of the possible role of RNAi and Dicer-like 1 in centromere function should be de-emphasized. In addition to the Mochizuki, K et al. (2004) paper cited for the role of Dcl1 in mitosis and meiosis, a second study, Malone, CD et al. (2005) examined the phenotypes of loss of Dcl1 function and observed no evidence for any role of Dcl1 in chromosome segregation. The differences between the two studies' findings were never reconciled so using only one study as supporting information for the Discussion may be misleading to the less informed reader.

15) This manuscript incorporates data from two other manuscripts. Namely the transposable element (TE) annotation (Kapusta et al.) and the deletion mapping (Cassidy-Hanley et al.), data used for chromosome landscape (Figure 2) and to delineate centromeric regions of the 5 germline chromosomes (in Figure 2 and Table 2). The TE paper seems to be "submitted" whereas the deletion mapping is "in preparation". I suggest that the deletion mapping be incorporated into this manuscript. Since this manuscript is somewhat wordy and speculative, it would not be difficult to tighten up the writing (shorten the manuscript) to make room for the deletion data, which is in fact already present except perhaps for an additional Table and/or Figure, and a paragraph in Materials and methods. This need not preclude a more detailed second paper on the subject. As for the transposable element annotation, I hope it can be published back-to-back with this manuscript. It would be even better (but probably not possible) to present at least the main results in the present manuscript, to gain a comprehensive overview of *Tetrahymena* germline chromosomes including not only CBS but also TE annotations.

16) The subsection "Scrambled Macronuclear Chromosome-Destined DNA" is weak. The word "scrambled" is misleading as the proposed mechanism (HR between repeats) is different than Oxytricha's use of a maternal guide RNA for unscrambling MDSs. The comparison with V(D)J joining is misleading, since that process involves the non-homologous end-joining (NHEJ) pathway and not HR. Finally, this is based on MAC DNA from a single karyonide followed by isolation of a single fully assorted vegetative clone, isn't it? How can such a rare occurrence process be validated without looking at DNA from independently rearranged and assorted lines?

17) How are IESs identified? Are they just operationally defined as regions in the MIC assembly that are deleted in the MAC assembly? Are there other criteria? It is mentioned in the fifth paragraph of the subsection “MIC Centromeres” that "we have characterized thousands of MIC-specific IESs". What is the characterization, just identification based on sequence alignment? Or is this a reference to data in the separate TE manuscript? I could not find anything in Materials and methods.

18) To avoid confusion, I think it should be made much clearer that IES in Paramecium are numerous, short, unique copy non-coding sequences (transposon remnants in many cases) while IES in *Tetrahymena* are (or include) the transposable elements and other repeated sequences (as clearly stated in the subsection “MIC Centromeres”). This should be made clear in the second paragraph of the Introduction (and why 40,000 instead of the published 45,000 Paramecium IESs?).

19) Since MIC and MAC telomere repeats are different, and the exact numbers of MIC and MAC chromosomes are known, could telomere repeat abundance (and Cbs abundance?) in the reads allow an estimate of MAC contamination (subsection “MIC Centromeres”, last paragraph)? If the MIC telomere repeats and previously characterized sub-telomeric regions can be identified in read pairs (in both short insert and jumping libraries), could they be mapped to the MIC scaffolds (subsection “Building and Analysis of Chromosome-Length MIC Super-Assemblies”, last paragraph)? With only 10 MIC chromosome ends (but 169X coverage), this might be difficult.

20) In the first sentence of the subsection “MIC Genome Sequencing and Assembly”, "highly AT-rich": can you provide the values for the MAC and MIC assemblies (77.7% and, 76.3%) or an average (~ 77%) in parentheses?

21) In the last sentence of the subsection “MIC Genome Sequencing and Assembly”, there is a typo ("as will *be* described"). More important, this sentence is not really justified, since only a small part of the Paramecium germline-limited sequences have been annotated and published to date. I don't see how a global comparison can be made, and it furthermore is made in the separate TE paper. That the organisation is different has been known for a long time, from molecular studies of the genome rearrangements.

22) In the sixth paragraph of the subsection “MIC Centromeres”. The suggestion that Paramecium MAC chromosomes are MIC chromosome arms was highly speculative when published in 2008, based on indirect evidence, namely an inverse correlation between G+C content and MAC chromosome length. It was suggested that this inverse correlation could be accounted for by biased gene conversion, indicating that MAC chromosomes could be proportional in size to MIC chromosome arms (since the MIC chromosomes are the ones undergoing meiotic recombination). More recently, work on Paramecium centromeres was published, so it would be wise to remove the speculative statement and refer only to the centromere paper. Moreover, I think that it is too soon to say that there is a fundamental difference in germline chromosome architecture at least as concerns centromeres, since it may turn out that centromeres are lost through each organism's usual IES elimination process.

23) Figure 3. Sequence logo. The Cbs logo does not present the frequency of each base, as stated in the legend. The logo is presented in the form of bits of information. However, this has not been calculated correctly. No A or T can carry 2 bits of information in a genome that is over75% A+T. The logo should be calculated taking into account the frequency of each base in the genome, instead of assuming equal frequencies. This will make the C's stand out a lot more. There is a "base frequency" view available using seqlogo, but the result is not as pretty.

24) In the fourth paragraph of the subsection “Conservation of chromosome breakage sites across *Tetrahymena* species”. I presume the term "centric chromosomes" refers to MAC chromosomes that come from centric regions of the MIC chromosomes. The text would be easier to follow if they are called centric MAC chromosomes. At the end of the same paragraph, is "decoupling" correct or should it be "uncoupling"?

25) Reference Lhuillier-Akakpo et al., 2015 is incomplete.

---

## [Author Response]

*[…] The manuscript has some remaining issues that need to be addressed before it would be considered as accepted, as outlined below. In particular, all three reviewers and the Reviewing Editor believe that the work will have its highest impact if it is more readable to general audience of eLife. The text body could have provided a bit more guidance for readers as to the general take-home implications of their findings, as detailed below. In addition, the manuscript would benefit from having a summary figure illustrating the major findings: repeat rich centromeric regions, conserved CBS positions, and non-maintained chromosomes.*

*1) Much of the information from the subsections “MIC Genome Sequencing and Assembly” and “Building and Analysis of Chromosome-Length MIC Super-Assemblies” could be stated in a summary fashion in the text and the details that are presented could be put in the Methods or in supplemental material. This would help the paper read more smoothly.*

As suggested by the reviewer, the detailed material in this section has been moved to the Materials and methods and a new Figure 1—figure supplement 1. We also moved details of genome sequencing and assembly to Materials and methods and the summary of assembly statistics (formerly Table 1) to [Supplementary-material SD5-data].

*2) The scrambled mac destined DNA described in the subsection “Scrambled Macronuclear Chromosome-Destined DNA” is of interest and the authors describe the mechanisms as different from what is in hypotrichs. It would be helpful to have a figure to follow what the authors have found. They state another manuscript will address this, but if they are going to report this at least to some degree here, a figure and a bit more detail about what is different would help a lot.*

We realized that making a comparison to gene unscrambling in Spirotrichs may be premature. We removed the separate heading of this section and moved a simplified paragraph, without reference to unscrambling, under the discussion of chromosome breakage. As a consequence, we believe it is no longer necessary to add another figure.

*3) The first six paragraphs of the subsection “MIC Centromeres” could be shortened and condensed. The whole section is particularly long and speculative, and in places repetitive. The concept of meiotic drive in inserted in the middle of the description of the structure and comparison to other organisms. Can the section be edited to tighten up the prose and perhaps shortened?*

The section on MIC centromeres has been considerably shortened. Some detailed information has been moved to a new Figure 2—figure supplement 1 and to [Supplementary-material SD5-data].

*4) The discussion of the programed elimination of Cen sequences in the sixth paragraph of the subsection “MIC Centromeres” needs to have a figure to allow the reader to follow the structures that are being referred to.*

The discussion of the programmed elimination of Cen sequences has been shortened and simplified; as a result, we do not believe it requires an additional figure.

*5) The seventh paragraph of the subsection “MIC Centromeres” discusses non-maintained chromosomes, which is discussed later. This does not belong under the heading "MIC Centromeres" where it currently is.*

The discussion of Non-Maintained Chromosomes has been moved to a later, more appropriate section of the manuscript.

*6) In the ninth paragraph of the subsection “MIC Centromeres”, the discussion of transcripts from mic sequences is really a summary of the literature and a hypothesis and should be in a dedicated discussion, not in the middle of a section on centromeres.*

In response to this point and point #14, this speculative hypothesis has been eliminated.

*7) The discussion of Cbs's in the subsections “The Chromosome Breakage Sequence (Cbs) Family”, “The Set of 225 Cbs’s in the MIC Genome” and “Degeneracy of the Cbs motif among functional Cbs’s”, should be significantly condensed. The consensus sequence was already known, so to authors should spend less time of this, and just start the new information that the additional sequences offer.*

As recommended, this section has been significantly condensed.

*8) Likewise the discussion of the conservation of CBS's across species in the subsections “Conservation of chromosome breakage sites across Tetrahymena species” and “Duplication of Cbs Regions on an Evolutionary Time Scale”, should be significantly shortened and combined with the identification of CSB sites discussed above.*

As suggested by the reviewer, we have significantly shortened the sections by moving details into the Materials and methods and supplementary files. We feel, however, that combining the identification and characterization of all 225 Cbs in *T. thermophila* with the discussion of Cbs conservation across species would not be appropriate. One of the main results of the former analysis is a complete picture of the range of Cbs sequence variability; the small sampling of Cbs’s identified in the other species fall within this range, as we point out in this section. However, the most interesting conclusion from the latter analysis is the conservation of Cbs locations, resulting in conservation of size and gene content of MAC chromosomes. This conclusion is entirely independent of the analysis of Cbs sequence conservation and would not be possible without the interspecies comparison in this section.

*9) The subsection “The Non-Maintained MAC Chromosomes” could be combined with the discussion of phenotypic assortments in the tenth paragraph of the subsection “MIC Centromeres” and this section should be shortened.*

In the interest of condensing the manuscript, the discussion of phenotypic assortment that the reviewer refers to has been moved to the legend of [Supplementary-material SD5-data]. We believe, in any case, that the discussion of non-maintained chromosomes stands on its own. As suggested, it has been shortened.

*10) The summary and conclusion would not be needed if the authors state the results and then wrote a concise Discussion.*

To avoid any repetition of results (and lengthening of the manuscript) that would be necessary if a separate Discussion section were included, we have included concise discussions that flow directly from the Results throughout the manuscript. We have greatly condensed and refocused the original “Summary and Conclusions” section, now titled “Conclusions and Future Directions”.

*11) On Figure 2, it seems like it might be helpful if the authors denoted the region which they consider to contain the centromere of each chromosome.*

The requested change has been made in the figure.

*12) In the fourth paragraph of the subsection “Duplication of Cbs Regions on an Evolutionary Time Scale”, it needs a period after "relatedness".*

Correction made.

*13) In the fourth paragraph of the subsection “Duplication of Cbs Regions on an Evolutionary Time Scale”. The idea about the CBS being introduced by a transposon invasion leading to the use of this sequence as the chromosome fragmentation site is compelling, but the way it is described is rather non-committal. It would not be "related" to a transposon invasion, it would result from the invasion and subsequent domestication of the breakage enzyme. This is one example related to my general comment that the manuscript could be written more succinctly and clearly.*

“Related to” has been changed to “resulted from”, as suggested.

*14) The description of the possible role of RNAi and Dicer-like 1 in centromere function should be de-emphasized. In addition to the Mochizuki, K et al. (2004) paper cited for the role of Dcl1 in mitosis and meiosis, a second study, Malone, CD et al. (2005) examined the phenotypes of loss of Dcl1 function and observed no evidence for any role of Dcl1 in chromosome segregation. The differences between the two studies' findings were never reconciled so using only one study as supporting information for the Discussion may be misleading to the less informed reader.*

This correction has been made. See point #6 above.

*15) This manuscript incorporates data from two other manuscripts. Namely the transposable element (TE) annotation (Kapusta et al.) and the deletion mapping (Cassidy-Hanley et al.), data used for chromosome landscape (Figure 2) and to delineate centromeric regions of the 5 germline chromosomes (in Figure 2 and Table 2). The TE paper seems to be "submitted" whereas the deletion mapping is "in preparation". I suggest that the deletion mapping be incorporated into this manuscript. Since this manuscript is somewhat wordy and speculative, it would not be difficult to tighten up the writing (shorten the manuscript) to make room for the deletion data, which is in fact already present except perhaps for an additional Table and/or Figure, and a paragraph in Materials and methods. This need not preclude a more detailed second paper on the subject. As for the transposable element annotation, I hope it can be published back-to-back with this manuscript. It would be even better (but probably not possible) to present at least the main results in the present manuscript, to gain a comprehensive overview of Tetrahymena germline chromosomes including not only CBS but also TE annotations.*

Two suggestions are made in this point. Publishing our analysis of *Tetrahymena* IES sequences back-to-back with the current manuscript, as suggested by the reviewer, was our original intent. The *eLife* editors decided not to fully review the IES manuscript, as originally submitted, but offered the option of moving “key results” into the first manuscript. Initially, we decided to move forward with full submission of the first manuscript, but after reading the reviews and consulting with Reviewing Editor Kathleen Collins, we decided to take the original advice and move key results on IES analysis into a comprehensive (but substantially condensed and thoroughly reorganized) manuscript on genome structure and rearrangement. The second suggestion was to incorporate the chromosomal deletion mapping data into this manuscript. The deletion mapping study has been a multi-year effort led Dr. Donna Cassidy-Hanley and involving many undergraduate researchers. Only a small subset of the deletions identified were informative about centromere locations. We are concerned that full publication of the methods and partial results from this effort would compromise the ability of Dr. Cassidy-Hanley and colleagues to publish the full story in a high quality journal and receive the attention and exposure that this story deserves. We do not feel that the omission of these detailed results detracts from our manuscript. However, we have included a new Figure 2—figure supplement 1 showing how a set of deletions were used to delimit the centromere region of Chromosome 5. The other centromeres were delimited in a similar fashion. We feel that this figure clarifies how the deletion analysis was performed, without compromising future publication of the entire study.

*16) The subsection "Scrambled Macronuclear Chromosome-Destined DNA" is weak. The word "scrambled" is misleading as the proposed mechanism (HR between repeats) is different than Oxytricha's use of a maternal guide RNA for unscrambling MDSs. The comparison with V(D)J joining is misleading, since that process involves the non-homologous end-joining (NHEJ) pathway and not HR. Finally, this is based on MAC DNA from a single karyonide followed by isolation of a single fully assorted vegetative clone, isn't it? How can such a rare occurrence process be validated without looking at DNA from independently rearranged and assorted lines?*

See our response to point #2.

*17) How are IESs identified? Are they just operationally defined as regions in the MIC assembly that are deleted in the MAC assembly? Are there other criteria? It is mentioned in the fifth paragraph of the subsection “MIC Centromeres” that "we have characterized thousands of MIC-specific IESs". What is the characterization, just identification based on sequence alignment? Or is this a reference to data in the separate TE manuscript? I could not find anything in Materials and methods.*

This information is now included. See our response to point #15.

*18) To avoid confusion, I think it should be made much clearer that IES in Paramecium are numerous, short, unique copy non-coding sequences (transposon remnants in many cases) while IES in Tetrahymena are (or include) the transposable elements and other repeated sequences (as clearly stated in the subsection “MIC Centromeres”). This should be made clear in the second paragraph of the Introduction (and why 40,000 instead of the published 45,000 Paramecium IESs?).*

The suggested corrections have been made.

*19) Since MIC and MAC telomere repeats are different, and the exact numbers of MIC and MAC chromosomes are known, could telomere repeat abundance (and Cbs abundance?) in the reads allow an estimate of MAC contamination (subsection “MIC Centromeres”, last paragraph)? If the MIC telomere repeats and previously characterized sub-telomeric regions can be identified in read pairs (in both short insert and jumping libraries), could they be mapped to the MIC scaffolds (subsection “Building and Analysis of Chromosome-Length MIC Super-Assemblies”, last paragraph)? With only 10 MIC chromosome ends (but 169X coverage), this might be difficult.*

We appreciate the suggestion of the reviewer to estimate the degree of MAC contamination by comparisons of MAC and MIC telomere repeats and to attempt to identify and link MIC telomeres to other scaffolds. We searched for MIC telomere repeats in the assemblies and reads, but unfortunately found few examples, and these uninformative. We have added text to the relevant section (Materials and methods in response to point #1) clarifying this issue. In any case, we do not feel that an estimate of MAC contamination would significantly add to or modify our conclusions. We clearly acknowledge that some contamination exists, which is unavoidable given experimental constraints, but as stated in the first paragraph of the Materials and methods, “By microscopic counting of purified nuclei (taking into account the relative nuclear ploidy), we estimate that contamination with macronuclear genomic DNA was less than 2%”. We are confident that this contamination did not significantly affect MIC genome assembly in any way that would affect the conclusions of this paper. Therefore, the text identified by the reviewer has been modified to explain the poor assembly of telomeric regions as being the result of their repetitive nature, as observed in the genome assemblies of many other organisms.

*20) In the first sentence of the subsection “MIC Genome Sequencing and Assembly”, "highly AT-rich": can you provide the values for the MAC and MIC assemblies (77.7% and, 76.3%) or an average (~ 77%) in parentheses?*

These values (MAC assembly 77.7% and MIC assembly 77.9%) have been added to the relevant section of the Materials and methods.

*21) In the last sentence of the subsection “MIC Genome Sequencing and Assembly”, there is a typo ("as will be described"). More important, this sentence is not really justified, since only a small part of the Paramecium germline-limited sequences have been annotated and published to date. I don't see how a global comparison can be made, and it furthermore is made in the separate TE paper. That the organisation is different has been known for a long time, from molecular studies of the genome rearrangements.*

The text in question has been removed.

*22) In the sixth paragraph of the subsection “MIC Centromeres”. The suggestion that Paramecium MAC chromosomes are MIC chromosome arms was highly speculative when published in 2008, based on indirect evidence, namely an inverse correlation between G+C content and MAC chromosome length. It was suggested that this inverse correlation could be accounted for by biased gene conversion, indicating that MAC chromosomes could be proportional in size to MIC chromosome arms (since the MIC chromosomes are the ones undergoing meiotic recombination). More recently, work on Paramecium centromeres was published, so it would be wise to remove the speculative statement and refer only to the centromere paper. Moreover, I think that it is too soon to say that there is a fundamental difference in germline chromosome architecture at least as concerns centromeres, since it may turn out that centromeres are lost through each organism's usual IES elimination process.*

We have modified the text to remove the speculative statement concerning *Paramecium* centromeres as well as the assertion of a “fundamental difference” between the centromeres of the two species and include reference to the recently published study, as suggested by the reviewer.

*23) Figure 3. Sequence logo. The Cbs logo does not present the frequency of each base, as stated in the legend. The logo is presented in the form of bits of information. However, this has not been calculated correctly. No A or T can carry 2 bits of information in a genome that is over75% A+T. The logo should be calculated taking into account the frequency of each base in the genome, instead of assuming equal frequencies. This will make the C's stand out a lot more. There is a "base frequency" view available using seqlogo, but the result is not as pretty.*

We thank the reviewer for catching the mistake in the figure legend; it has been corrected to identify the y-axis as bits, that is, as conservation units rather than nucleotide frequency units. Regarding the suggestion to raise the baseline to the level of the genome-wide conservation level, which is what the reviewer suggests, there is some merit to the suggestion. But the genome-wide frequencies (and associated bits of conservation) include coding sequence frequencies that have higher than average G+C content. Retaining the true conservation level of the non-coding Cbs-adjacent sequence provides a meaningful internal control. We prefer to retain the presentation of the conservation measurements in an assumption-free way, which does not, for example, give a distorted view of the difference in conservation between the T’s at Cbs position 1 and 15.

*24) In the fourth paragraph of the subsection “Conservation of chromosome breakage sites across Tetrahymena species”. I presume the term "centric chromosomes" refers to MAC chromosomes that come from centric regions of the MIC chromosomes. The text would be easier to follow if they are called centric MAC chromosomes. At the end of the same paragraph, is "decoupling" correct or should it be "uncoupling"?*

Following revision, the term "centric chromosomes" no longer appears in the main text, but it has been replaced with "centric MAC chromosomes" in the legend to Table 1. In the legend to [Supplementary-material SD5-data], we also now refer to "Centric MAC chromosomes, derived from the centromeric regions of MIC chromosomes". The suggested change from "decoupling" to "uncoupling" has been made. This sentence now appears in [Supplementary-material SD6-data].

*25) Reference Lhuillier-Akakpo et al., 2015 is incomplete.*

The reference has been corrected.